# Pharmacological and non-pharmacological methods of inducing wakefulness activate distinct neural populations in the mouse brain

Renato Maciel[1☉], Justin Malcey[1☉], Kylian Gautier[1☉], Amarine Chancel[1], Blandine Duval[1], Théo Brunel[1], Patrice Fort[1], Claudio Marcos Queiroz[2‡], Pierre-Hervé Luppi[1‡*]

1 INSERM, U1028; CNRS, UMR5292; Lyon Neuroscience Research Center, Team "Physiopathologie des réseaux neuronaux responsables du cycle veille-sommeil", Lyon, France, University Lyon 1, Lyon, France, 2 Brain Institute, Universidade Federal do Rio Grande do Norte, Natal, Brazil

☉ These authors contributed equally to this work.
‡ CMQ and P-HL also contributed equally to this work.
* luppi@sommeil.univ-lyon1.fr

## Abstract

A large body of data indicate that the aminergic, cholinergic and hypocretin/orexin neurons are responsible for inducing wakefulness. However, recent data showed that other systems might also play a key role. Further, wakefulness induced by different drugs versus non-pharmacological means could be generated by different populations of neurons. To address these questions, we evaluated at the whole brain level in the same mice using TRAP2 model whether the same neurons were activated by the wake-inducing drugs modafinil and solriamfetol versus non-pharmacological wake. Our results show that nine subcortical structures namely the oval part of the bed nucleus of the stria terminalis, lateral part of the central amygdalar nucleus, paraventricular hypothalamic and thalamic and supraoptic nuclei, external part of the lateral parabrachial nucleus, caudal part of the nucleus of the solitary tract and the area postrema are significantly more activated by solriamfetol than modafinil and non-pharmacological wakefulness. In contrast, a second category of structures including the orexin neurons, the parasubthalamic and laterodorsal tegmental nucleus are strongly activated in all types of induced wake. Further, some classical wake systems like the dopaminergic neurons of the ventral tegmental area or the dorsal raphe nucleus and the noradrenergic neurons of the locus coeruleus are either very poorly or not strongly activated. These results reveal that many structures not previously involved in wakefulness might play a key role in regulating the state and that some structures might be more recruited by solriamfetol than modafinil or non-pharmacological wakefulness. Our results are particularly relevant for pathologies such as hypersomnia. They open a new era in the study of the mechanisms responsible for inducing wakefulness.

**Data availability statement:** All relevant data are within the paper and its Supporting information files.

**Funding:** This work was supported by CNRS (UMR5292)(to PHL), INSERM (U1028) (to PHL), SFRMS)(to AC, BD), University Lyon 1 (to PHL), Conselho Nacional de Desenvolvimento Científico e Tecnológico (CNPq, grants numbers 308110/2020-0 and 316441/2023-6) (to CMQ) and Fundação Coordenação de Aperfeiçoamento de Pessoal de Nível Superior (CAPES, grant number #385/2019, CAPES-COFECUB) (to CMQ and PHL). The study was funded by an unrestricted grant from Jazz Pharmaceutical and then Axsome and Pharmanovia (to PHL). This work was performed within the framework of the LABEX CORTEX (ANR-11-LABX-0042) of l'Université Claude Bernard Lyon 1, within the program "Investissements d'Avenir" (decision n° 2019- ANR-LABX-02) operated by the French National Research Agency (ANR)(to PHL). The funders had no role in study design, data collection and analysis, decision to publish, or preparation of the manuscript.

**Competing interests:** We have read the journal's policy and the authors of this manuscript have the following competing interests: PHL is a member of PLOS Biology's Editorial Board.

**Abbreviations:** μ, mean; 4-OHT, 4-hydroxytamoxifen; 5HT, serotonin; ACB, nucleus accumbens; Ach, acetylcholine; ANOVA, analysis of variance; BNST, bed nucleus of the stria terminalis; CEA, central amygdalar nucleus; CGRP, calcitonin gene-related peptide; CI, confidence interval; CRH, corticotropin releasing hormone; CTX, cortex; DA, dopaminergic; DAPI, 4′,6-diamidino-2-phenylindole; DMSO, dimethyl sulfoxide; DRN, dorsal raphe nucleus; EEG, electroencephalogram; EMG, electromyogram; FWER, family-wise error rate; GABA, gamma-aminobutyric acid; Glu, glutamate; Hcrt, hypocretin; HDC, histidine decarboxylase; HF, hippocampal formation; Hist, Histamine; HSD, Tukey Honestly Significant Difference test; HYP, hypothalamus; IP, intraperitoneal; LC, locus coeruleus; LDT, laterodorsal tegmental nucleus; LHA, lateral hypothalamic area; LPB, lateral parabrachial nucleus; LRN, lateral reticular

## Introduction

Wakefulness is defined by electroencephalogram (EEG) activation, muscle activity, and eye movements. It is well accepted that multiple redundant subcortical structures are responsible for inducing wakefulness [1,2,3,4]. These systems correspond to the aminergic (histaminergic, catecholaminergic, serotonergic), cholinergic, and the hypocretin/orexin hypothalamic neurons. Although additional populations of wake-inducing neurons have been described recently [5,6,7,4], their role has not been confirmed. The aim of the present study was to examine over the whole brain for all systems potentially inducing wakefulness. Further, we aimed to determine whether wakefulness induced by different drugs such as modafinil (Mod) or solriamfetol (Sol) or by sensory stimulation (non-pharmacological wakefulness, NW) is induced by the same systems. Solriamfetol and modafinil are norepinephrine–dopamine reuptake inhibitors (NDRI) used for the treatment of excessive sleepiness associated with narcolepsy and sleep apnea [8,9,10]. Both drugs have been shown to induce wakefulness in mice without the side effects obtained with amphetamine [11]. Based on the mode of action of solriamfetol and modafinil, it can be hypothesized that wakefulness induction is due to increased concentrations of norepinephrine and dopamine in the synapses [12]. Such increment would lead to enhanced activation of wake-inducing neurons expressing the receptors of one or two monoamines. Most neurons in the brain express dopamine and/or norepinephrine receptors. Therefore, determining which population(s) of neurons are involved in the induction of waking by solriamfetol and modafinil is challenging. In addition, it has recently been shown that Sol but not Mod has additional agonist activity at the trace amine associated receptor 1 (TAAR1) [13] and TAAR1 agonists induces W and inhibits NREM sleep [14]. It is therefore possible that the two drugs activate only partly the same wake-inducing structures.

One strategy to identify the neurons responsible for inducing wakefulness would be to determine the effect of solriamfetol, modafinil and NW on neurons already known to be involved in the induction of the state. Another strategy is to identify neurons activated by solriamfetol, modafinil and NW across the entire brain without a priori using the cFos method. Our laboratory has been successfully using the cFos method for more than two decades to identify the neuronal network responsible for inducing paradoxical (REM, Rapid Eye Movement) sleep (PS) and to compare it with wakefulness. Using such a method, we have been the first to identify the neuronal network responsible for inducing muscle atonia during PS and to show that melanin-concentrating hormone neurons (MCH) play a key role in PS generation [15,16,17]. Further, we showed that cortical activation during PS is restricted to a few limbic cortical structures compared to wakefulness opening the avenue to the identification of the function of the state [18].

Hasan and colleagues [11] qualitatively compared cFos staining obtained after wakefulness induced by solriamfetol, modafinil, and amphetamine. Their limited analysis of cFos distribution indicated that the three drugs activate different populations of neurons [11]. However, it was not detailed enough to identify the populations of neurons likely responsible for the induction of wakefulness. Further, the cFos method

nucleus; LS, lateral septum; MCH, melanin-concentrating hormone; Md, median; MED, medulla; MES, mesencephalon; Mo, mode; Mod, Modafinil; MPO, medial preoptic nucleus; NA, noradrenergic; NDRI, norepinephrine–dopamine reuptake inhibitors; NTS, nucleus of the solitary tract; NREM, non-rapid eye movement; NW, non-pharmacological wakefulness; Orx, orexin; OT, oxytocin; PBS, Phosphate-Buffered Saline; PBST, Phosphate-Buffered Saline Tween; PCA, Principal Component Analysis; PON, pons; PS, paradoxical sleep; PSTN, parasubthalamic nucleus; PVN, paraventricular hypothalamic nucleus; PVT, paraventricular thalamic nucleus; REM, Rapid Eye Movement; RI, Reactivation Index; SEM, standard error of the mean; SN, substantia nigra; SON, supraoptic nucleus; SOI, structures of interest; Sol, Solriamfetol; SSp, somatosensory; SWS, slow waves sleep; TAAR1, trace amine associated receptor 1; tdT, tdTomato; TEL, telencephalon; TH, tyrosine hydroxylase; THA, thalamus; TMN, tuberomammillary nucleus; TRAP2, targeted recombination in active populations; VLPO, ventrolateral preoptic area; VP, vasopressin; VTA, ventral tegmental area; W, wakefulness; ZI, zona incerta.

does not allow to determine whether the same or different neurons are activated by different drugs or by wakefulness induced by non-pharmaceutical means.

To reach this objective, we used in the present study an updated genetic method of cFos allowing us to determine in the same mice whether the same neurons are activated during two successive periods of wakefulness. We recently used this method based on a new type of transgenic (TRAP) mice [19,20,21]. Using such an innovative model, we did determine whether the same or different neurons are activated by solriamfetol and modafinil or by wakefulness induced by non-pharmaceutical intervention. To this aim, we induced in the same mice one week apart two periods of wakefulness, one using solriamfetol and the other one with modafinil or by gently stimulating the mice when they were falling asleep. The expression of the reporter gene tdTomato was induced in the activated neurons by injecting 4-hydroxytamoxifen (4-OHT) during the first period of wakefulness. One week later, a second period of wakefulness was induced by another means and the mice were perfused. A whole brain mapping was then made to localize neurons expressing cFos and/or tdTomato. To our knowledge, our report is the first to compare the cFos and the tdTomato staining in the TRAP2 mice over the entire brain. Further, such mapping allowed us to identify neurons activated during wakefulness induced by solriamfetol versus modafinil or non-pharmaceutical induced wakefulness.

## Results

### Behavioral and electrophysiological analysis

To comprehensively assess the wake-promoting actions of solriamfetol and modafinil, we conducted two-hour video-EEG recordings after each treatment and compared these profiles with mice undergoing non-pharmacological wakefulness induced by sensory stimulation, during the light (NWday) or the dark (NWnight) period (Fig 1). Computer-assisted animal tracking revealed clear differences in ambulation patterns across treatments and sessions (Fig 2). In the both sessions, solriamfetol-treated mice showed significantly decreased locomotion, with heat maps indicating predominantly peripheral paths and limited exploration. By contrast, modafinil and NW groups displayed more extensive trajectories, covering larger portions of the arena and showing higher occupancy in central zones (Fig 2A). Velocity traces reflected these occupancy maps, with solriamfetol-treated animals moving less and presenting more immobile epochs than modafinil-treated and NW mice (Fig 2B). Quantification of total distance traveled confirmed these differences, showing that modafinil and NW significantly increased ambulation relative to solriamfetol (Fig 2C; $p < 0.05$ for both first and second sessions). These results show that non-pharmacological wakefulness and modafinil promote waking with robust locomotor activation, whereas solriamfetol produces waking with subtle ambulatory response in both sessions.

To determine whether the lower ambulatory activity observed in solriamfetol-treated mice reflected reduced arousal or simply diminished motor output, we analyzed EEG spectral properties across during the same two-hour recording session (Fig 3). Time–frequency spectrograms revealed that all groups (solriamfetol, modafinil, and non pharmacological wakefulness) displayed brain oscillations and

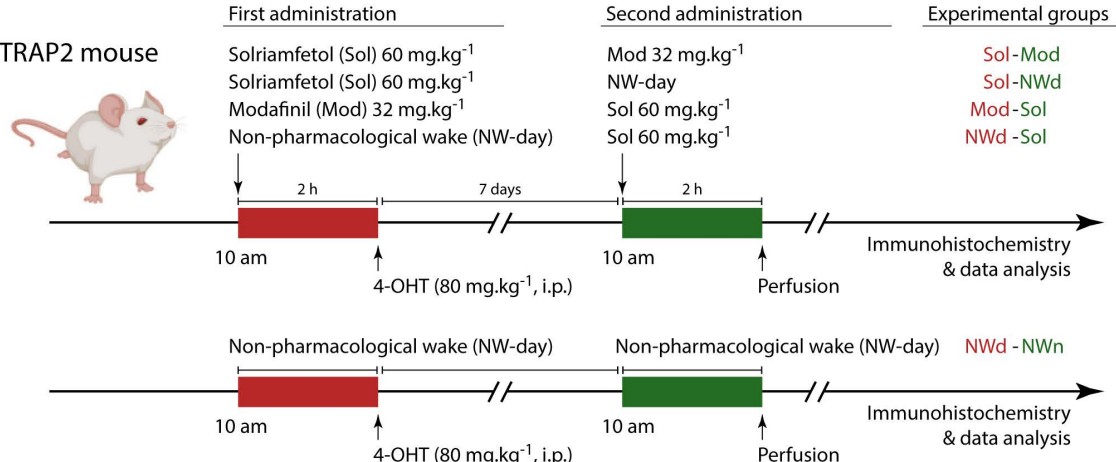

**Fig 1. Experimental design.** Male adult (8-12 weeks old mice) TRAP2 mice were placed in the recording chamber for seven days before the beginning of the experiments. On the first experimental day, animals received an IP injection of solriamfetol (Sol) or modafinil (Mod) at 10 AM The introduction of new objects in the recording chamber and occasional sensory stimulation was used to induce non-pharmacological wake (NW). All animals were injected with 80 mg/kg of 4-OHT 2 hours after the administration of the experimental drugs or wake induction to induce the expression of the reporter gene (tdTomato) in cFos-expressing neurons. Seven days after the first condition, animals returned to the recording chamber and received an administration of the other drug or were subjected to NW. Animals were perfused 2 hours later and their brains were processed for immunofluorescence staining of cFos. Each group (top-right) contained four animals (16 animals in total).

elevated EMG tone associated with wakefulness most of the two-hour session time (Fig 3A). Sleep scoring revealed that all animals remained continuously awake in the first hour after injections (S1A Fig). In the second hour, wake time started to decrease in particular for modafinil-treated mice for the first session ($H[2,13] = 6.40$, $p = 0.041$, third 30-min interval; $H[2,13] = 8.31$, $p = 0.016$, fourth 30-min interval; Kruskal–Wallis test) (S1A Fig). In the second experimental session, a significant decrease in wake time was observed for both Sol- and Mod-treated groups, but only during the fourth 30-min interval ($H[2,13] = 7.48$, $p = 0.024$, Kruskal–Wallis test, S1B Fig).

Average power spectral density across the two-hour session revealed an interesting waking profile regarding the expected dominance of theta and delta frequency bands (Fig 3B). As shown in Fig 3A, solriamfetol treatment slowly decreased theta peak frequency. In this representative example, theta peak frequency starts slowing down just after drug administration reaching its minimum value 90–100 min after treatment. Therefore, we computed the power and peak frequency of delta and theta oscillation in three 10-min long epochs, immediately after drug administration (0–10 min), and in the middle (between 55–65 min) and end (110–120 min) of the recording session (Fig 3C and 3D). Delta power and delta peak frequency showed no significant differences between groups across any epoch (Fig 3C), indicating that none of the treatments promoted sustained slow wave oscillations, commonly associated with slow wave sleep. In contrast, theta-band analyses revealed modest but systematic group differences. Theta power was comparable across treatments, whereas the theta peak frequency was significantly lower in the solriamfetol or modafinil groups during the second and third epochs compared with the NW condition (Fig 3D). Importantly, despite these subtle spectral shifts, all groups maintained a clear waking EEG profile in every epoch. Together, these results demonstrate that the reduced locomotion observed under solriamfetol is not attributable to sleep intrusions but reflects a wake state with lower motor activity and theta peak frequency.

## Overall analysis of neuronal activation

To determine the structures differentially activated by Sol, Mod, NWday, and NWnight as well as those strongly activated during all conditions, the number of tdT, cFos, and double-labeled neurons was quantified in the whole brain in

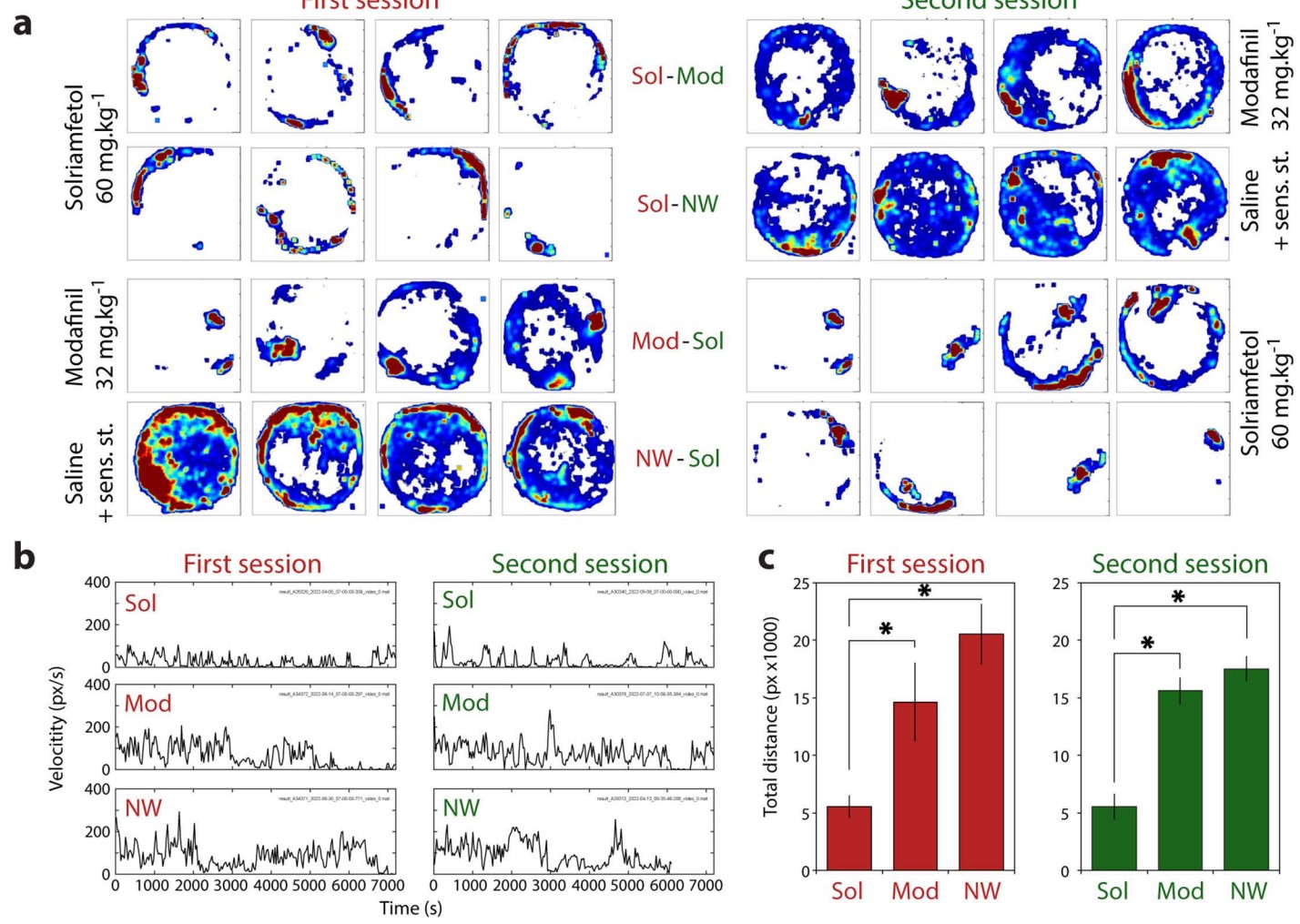

**Fig 2. Spontaneous locomotor activity after experimental treatment. (a)** Occupancy maps during the two-hour recording period following administration of the experimental drugs or saline, shown for both recording sessions and all animals. Each square represents one animal; warm colors indicate longer time spent in that arena location. **(b)** Representative locomotor traces over the two-hour recording session (7,200 s) for Sol-, Mod-, and saline-treated animals in both sessions. **(c)** Total distance traveled (in pixels) across the two recording sessions. Data are presented as mean ± SD. *$p < 0.05$ (unpaired t test). Note the significant reduction in locomotor activity in Sol-treated animals. Raw data underlying the Figure is shown in S1 Data.

all mice (10 sections for every 21 mice). We evaluated 215 structures of interest (SOI) without a priori assumptions, of which 112 structures met the inclusion criteria (values from at least three animals per group), organized in eight macroregions: cortex, telencephalon, thalamus, hypothalamus, hippocampal formation, mesencephalon, pons, and medulla. A total of 1,385,931 tdT+ cells and 1,568,182 cFos+ cells were counted in all mice, and no statistical difference was observed between the number of tdT and cFos positive cells (mean ± SEM per mouse, tdT: 65,997 ± 7,821 and cFos: 74,675 ± 9,098, p = 0.10, paired t test). Moreover, the expression of tdT and cFos was strongly correlated in the structures analyzed in all animals ($R^2 = 0.80 ± 0.02$, mean ± SEM, N = 16; $p < 10^{-21}$, Pearson correlation on log10 transformed number of tdT and cFos positive cells; Fig 4). Similar correlations were observed for NWday-NWnight animals (S2 Fig).

We then calculated the density of tdT and cFos-labeled neurons across brain regions and individual structures and found an overall similar pattern in cell distribution for the two markers (Fig 5). The highest cell density was found in the

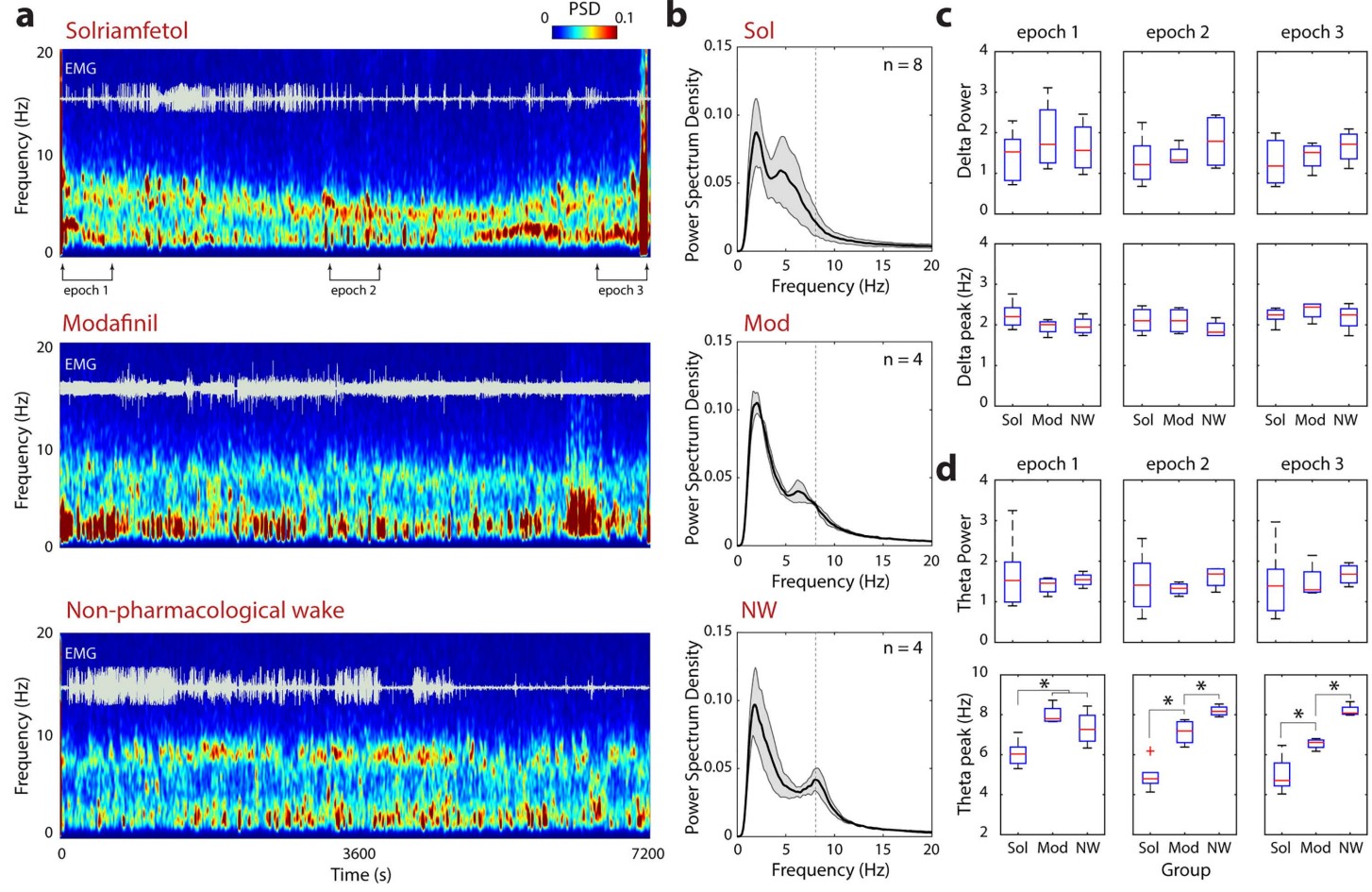

**Fig 3. Spectral characteristics of wakefulness induced by solriamfetol and modafinil compared to non-pharmacological wake. (a)** Representative 2-hour long EEG spectrograms (0–20 Hz) with corresponding EMG signal (white traces) in three experimental groups: solriamfetol (top), modafinil (middle), or saline (natural waking; bottom). Warmer colors indicate higher power spectral density (PSD). Horizontal bars denote the three epochs used in power quantification. **(b)** Averaged 2-hour long power spectral density during wakefulness induced by solriamfetol (Sol; $n=8$), modafinil (Mod; $n=4$), and natural waking (NW; $n=4$). Data show mean (black tick line) and SEM (gray shade). Dashed vertical lines mark the theta peak frequency in the NW group. **(c)** Delta power and delta peak frequency across epochs 1–3 for Sol, Mod, and NW groups. Boxplots show median (red line), interquartile range (box), and full range (whiskers). **(d)** Theta power and theta peak frequency across epochs 1–3. Significant statistical differences between groups are indicated (*$p<0.05$). Raw data underlying the Figure is shown in S2 Data.

hypothalamus, in the paraventricular hypothalamic nucleus (PVN) and the supraoptic nucleus (SON), while the hippocampal formation (HF) showed the lowest cell density across all brain regions (Fig 5).

The values of the reactivation index (RI) corresponding to the number of tdT-cFos double-labeled neurons divided by the total number of tdT neurons from 113 structures and 21 animals are shown in Fig 6. The histogram is well described by a gamma distribution ($R^2=0.97$, $p=4.3 \times 10^{-7}$) with average RI of 0.24 and median RI of 0.22 (considering all measurements). Irrespective of the experimental condition, the average RI was higher in the mesencephalon (RI = $0.292 \pm 0.008$), hypothalamus (RI = $0.285 \pm 0.008$), pons (RI = $0.275 \pm 0.011$), and medulla (RI = $0.255 \pm 0.007$) compared with cortex (RI = $0.206 \pm 0.005$), telencephalon (RI = $0.223 \pm 0.006$), and thalamus (RI = $0.203 \pm 0.009$; $F_{[7,1952]}$ = 25.82, $p=5.6 \times 10^{-34}$, one-way ANOVA). The hippocampal formation showed the lowest RI among macrostructures (RI = $0.077 \pm 0.005$).

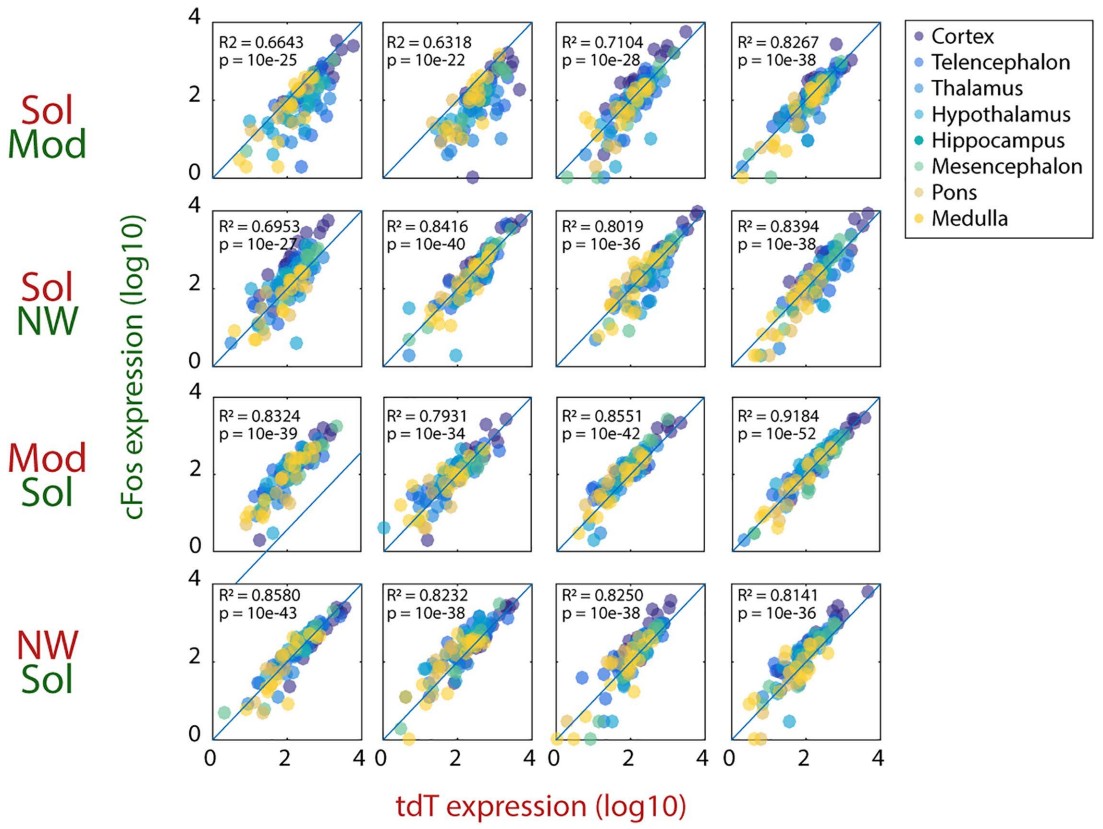

**Fig 4. Correlation between the expression of tdTomato and cFos.** The log transforms numbers of tdT and cFos labeled neurons per structure are linearly correlated in the four animals per Sol-Mod, Mod-Sol, Sol-Nwday and Nwday-Sol group (top to bottom, Pearson correlation, R2 and *p*-value shown in each panel). Note that most of the structures are closed to the diagonal line. Raw data underlying the Figure is shown in S3 Data.

## Structures differentially activated by Sol, Mod, or NW

**Identification of the structures specifically activated by Sol.** We defined a structure as more activated in the Sol condition compared to the three other conditions when tdT neuron density was higher in Sol than in Mod and NWday conditions, cFos density was higher in Sol than in Mod, NWday, and NWnight conditions and the reactivation index (RI = cFostdt/tdt) was higher in Sol–Mod than in Mod–Sol groups and in the Sol-NWday than in the NWday-Sol group. Finally, of 9 structures was characterized by a significantly increased density of tdT and cFos activated cells in the Sol condition compared to the Mod and NWday and night conditions (S1 Table) combined with a RI superior in the Sol-Mod and the Sol-NW groups compared to the mirror groups (S2 Table). This group was composed of the bed nucleus of the stria terminalis as a whole and of the oval nucleus of its anterior division (Fig 7B and 7D), the lateral part of the central amygdalar nucleus (Fig 7C and 7E), paraventricular hypothalamic (Fig 8B and 8D) and thalamic nuclei (Fig 9B and 9D), supraoptic nucleus (Fig 8), external part of the lateral parabrachial nucleus (Fig 9C and 9E), caudal part of the nucleus of the solitary tract (Fig 10B and 10D) and the area postrema (S1 and S2 Tables).

Two limbic structures were included in the selection. The bed nucleus of the stria terminalis was statistically more activated in the Sol than in the Mod, NWday and NWnight conditions. Importantly, the large majority of the neurons specifically activated in the Sol condition were localized in its anterior division, lateral part (Fig 7B and 7D) (S1 Table). The second limbic structure with an increased activation specifically in the Sol condition was the lateral part

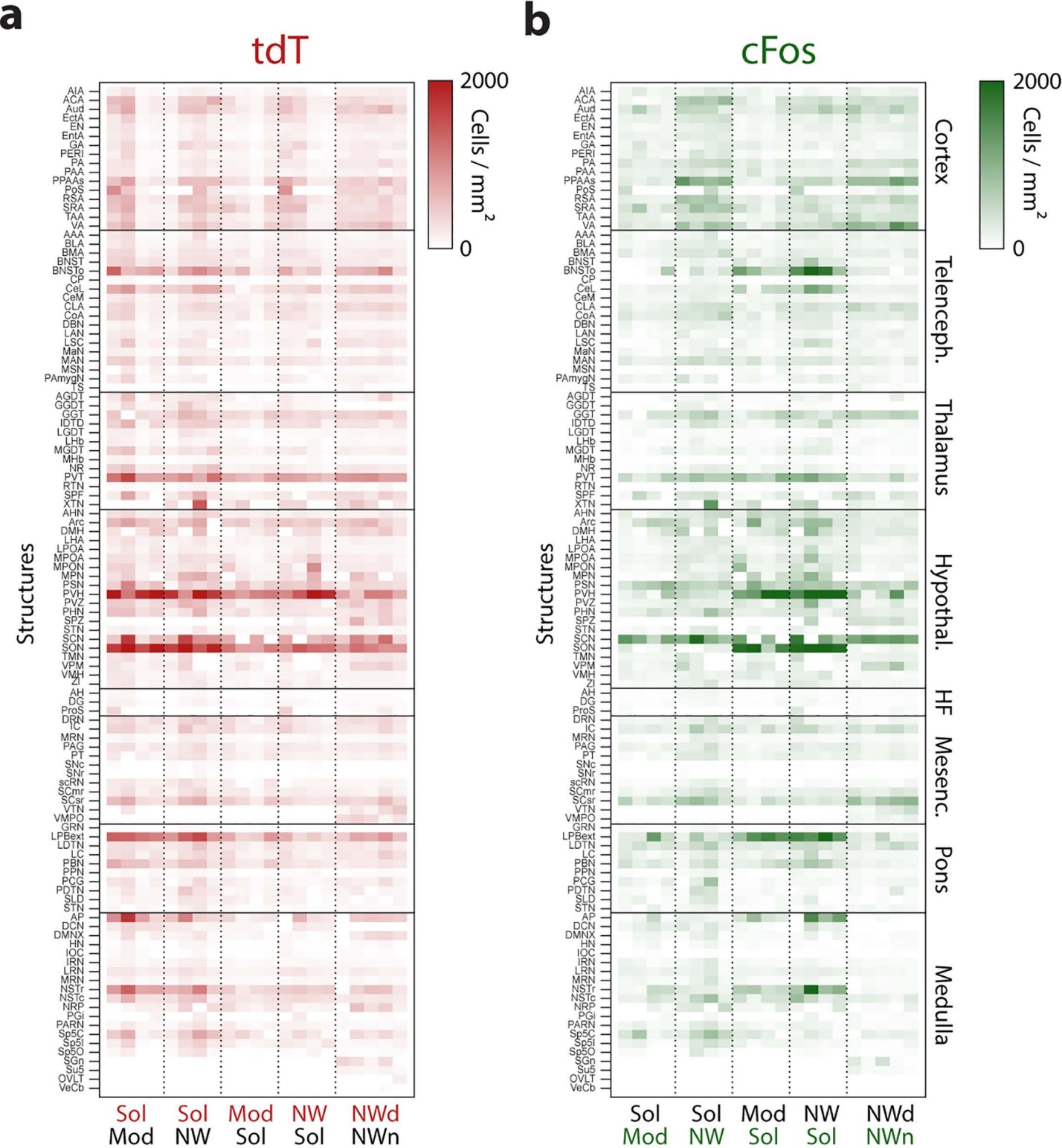

**Fig 5. Brain-wide expression of tdTomato (tdT) and cFos after solriamfetol (Sol), modafinil (Mod), and non-pharmacological wakefulness day and night (Nwday, Nwnight).** Cell density heatmap representation of tdTomato (a, left) and cFos (b, right) in the cortex, telencephalon, thalamus, hypothalamus, hippocampal formation (HF), mesencephalon, pons, and medulla. Each column represents one mouse. The cell density is color coded from zero to 23,000 cells/mm². Raw data underlying the Figure is shown in S3 Data.

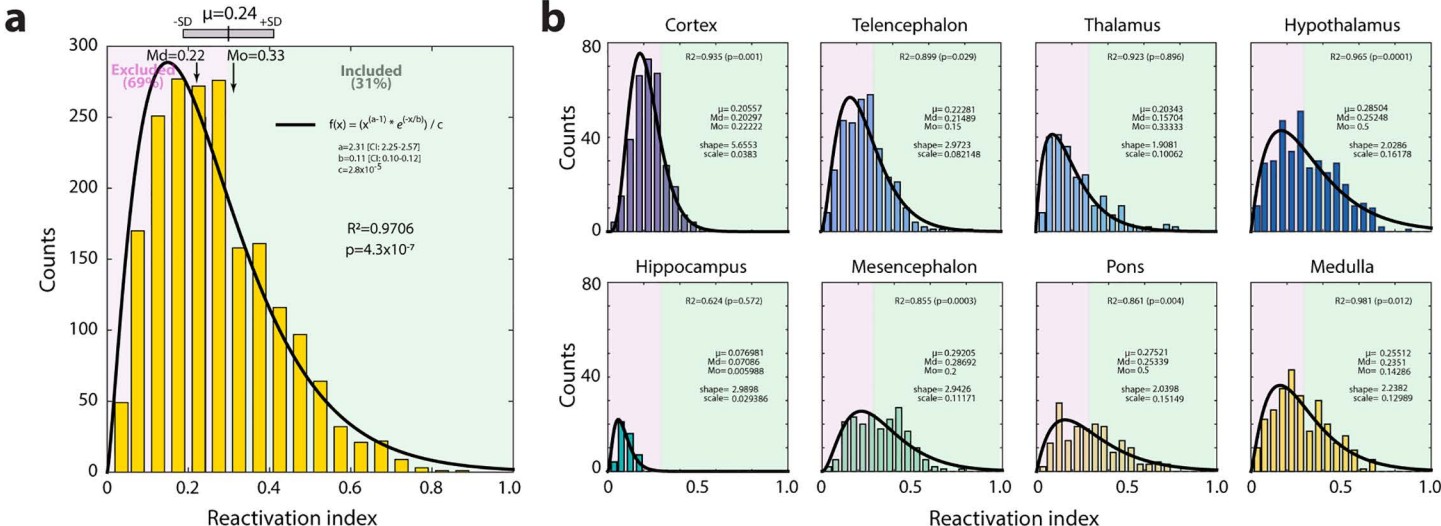

**Fig 6. Distribution of the reactivation index (RI). (a)** Histogram of the RI for 113 structures and 21 animals (bin size = 0.05). The histogram was fitted to a gamma distribution with parameters shape ($a = 2.31$) and scale ($b = 0.11$; $R2 = 0.97$, $p = 4.3 \times 10^{-7}$). The mean ($\mu$), median (Md) and mode (Mo) is shown in the Fig, together with the parameters (CI: confidence interval), R2 and $p$-value. **(b)** Histograms of the RI for each macroregion. A threshold above 0.3 in the four groups was set as an inclusion criteria for populations of neurons activated during all types of wakefulness. Raw data underlying the Figure is shown in S3 Data.

of the central amygdala (Fig 7C and 7E) (S1 Table). For these two structures, the RI was significantly above in the Mod-Sol and NWday-Sol mice groups compared to the mirror groups (Fig 7) (S2 Table). Importantly, the medial part of the central amygdala did not show a differential activation between Sol, Mod, NWday, and NWnight conditions (Fig 7).

Two hypothalamic structures known to contain oxytocin and vasopressin neurons projecting to the neurohypophysis displayed the highest density of cFos and tdT neurons in the Sol condition among the 9 Sol-specific structures (Fig 8) (S1 Table). For these two structures, more than 43% of the tdT neurons were reactivated in the Mod-Sol and NWday-Sol mice groups compared to less than 9% in the mirror groups (S2 Table).

One thalamic structure namely the paraventricular nucleus of the thalamus was significantly more activated in the Sol than in the other conditions, but contrary to the 8 other structures, the density of cFos and tdt neurons remained high in the other conditions (Fig 9B and 9D) (S1 Table). In agreement with these results, its RI was statistically superior in the Mod-Sol mice compared to the Sol-Mod mice, but it did not reach statistical difference between the NWday-Sol mice and the Sol-Nwday mice.

One pontine structure, namely the external part of the lateral parabrachial nucleus showed a strong and significant increased activation in the Sol condition compared to the Mod and Nwday and Nwnight conditions (Fig 9C and 9E) (S1 and S2 Tables). The difference was specifically observed in this subdivision of the nucleus and not in the other parts of the lateral parabrachial nucleus (Fig 9C). For this structure, the percentage of Tdt neurons reactivated was above 46% in the Mod-Sol and Nwday-Sol groups while it was below 17% in the Sol-Mod and Nwday-Sol groups confirming that most neurons activated in the Sol condition were not activated in the other conditions (S2 Table).

Finally, two closely located and functionally linked medullary structures, the caudal part of the nucleus of the solitary tract and the area postrema were significantly more activated in the Sol condition compared to the three other conditions (Fig 10B and 10D) (S1 Table). In the caudal part of the nucleus of the solitary tract, more than 43% of the Tdt neurons were reactivated in the Mod-Sol and NWday-Sol groups while less than 13% of them were reactivated in the Sol-Mod and

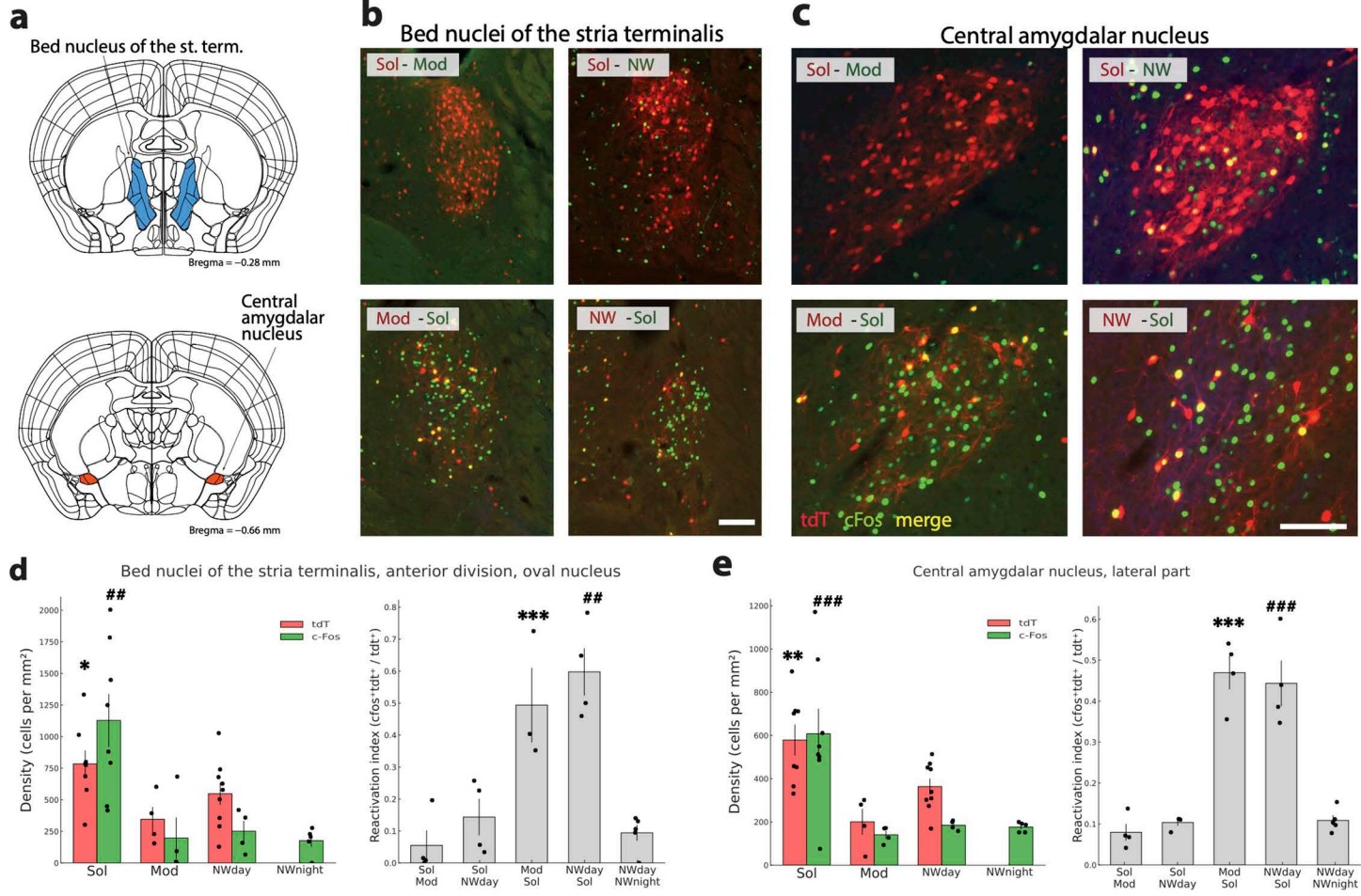

**Fig 7. Limbic structures more activated by Solriamfetol. (a)** Drawings showing the localization of the two structures **(b, c)** Photomicrographs illustrating the distribution of tdT (red), c-Fos (green) and double-labeled neurons (yellow) in one representative mouse per experimental group in the bed nucleus of the stria terminalis and central amygdalar nucleus. **(d, e)** Histograms (left) showing the mean ± sem density (cell/mm²) of tdT+ (red) and cFos+(green) neurons with each mice (dots) also displayed for the four conditions (Sol, Mod, Nwday, Nwnight). Note that for the oval nucleus of the anterior division of the bed nucleus of the stria terminalis and the lateral part of the central amygdala, the density of cFos and tdT neurons is significantly higher in the Sol condition compared to the Mod, Nwday and Nwnight conditions. Histograms on the right (gray) show the reactivation index (RI) in each group of mice (dots show each mice value). Note that the RI is significantly higher in the Mod-Sol and Nwday-Sol groups compared to their respective mirror groups. Significance tested with generalized linear models (Gamma family, log link) using robust (HC0) standard errors, one-sided in the "greater" direction with the animal as the experimental unit. Significantly different tdT density in the Sol condition vs. the other conditions: $p < 0.05$ * $p < 0.01$ ** $p < 0.001$ ***. Significantly different cFos density in the Sol condition vs. the other conditions: $p < 0.05$ #, $p < 0.01$ ##, $p < 0.001$ ###. Significantly different RI between Mod-Sol and Sol-Mod mice $p < 0.05$ * $p < 0.01$ ** $p < 0.001$ ***. Significantly different RI between Nwday-Sol and Sol-Nwday mice $p < 0.05$ #, $p < 0.01$ ##, $p < 0.001$ ###. Scale b,c: 100 μm. Raw data underlying the Figure is shown in S3 Data.

Sol-Nwday groups indicating that most neurons from these structures are activated specifically during Sol (Fig 10D) (S2 Table). The percentage of tdT neurons reactivated in the Mod-Sol and Nwday-Sol conditions was above 21% while it was below 9% in the Sol-Mod and Sol-Nwday conditions confirming that most area postrema neurons were specifically activated during Sol. Importantly, the rostral part of the nucleus of the solitary tract was highly activated in all conditions (see below).

**Identification of the structures specifically activated in the Mod, Nwday, or Nwnight conditions.** We defined a structure as more activated in the Mod condition compared to the three other ones when tdT neuron density was higher

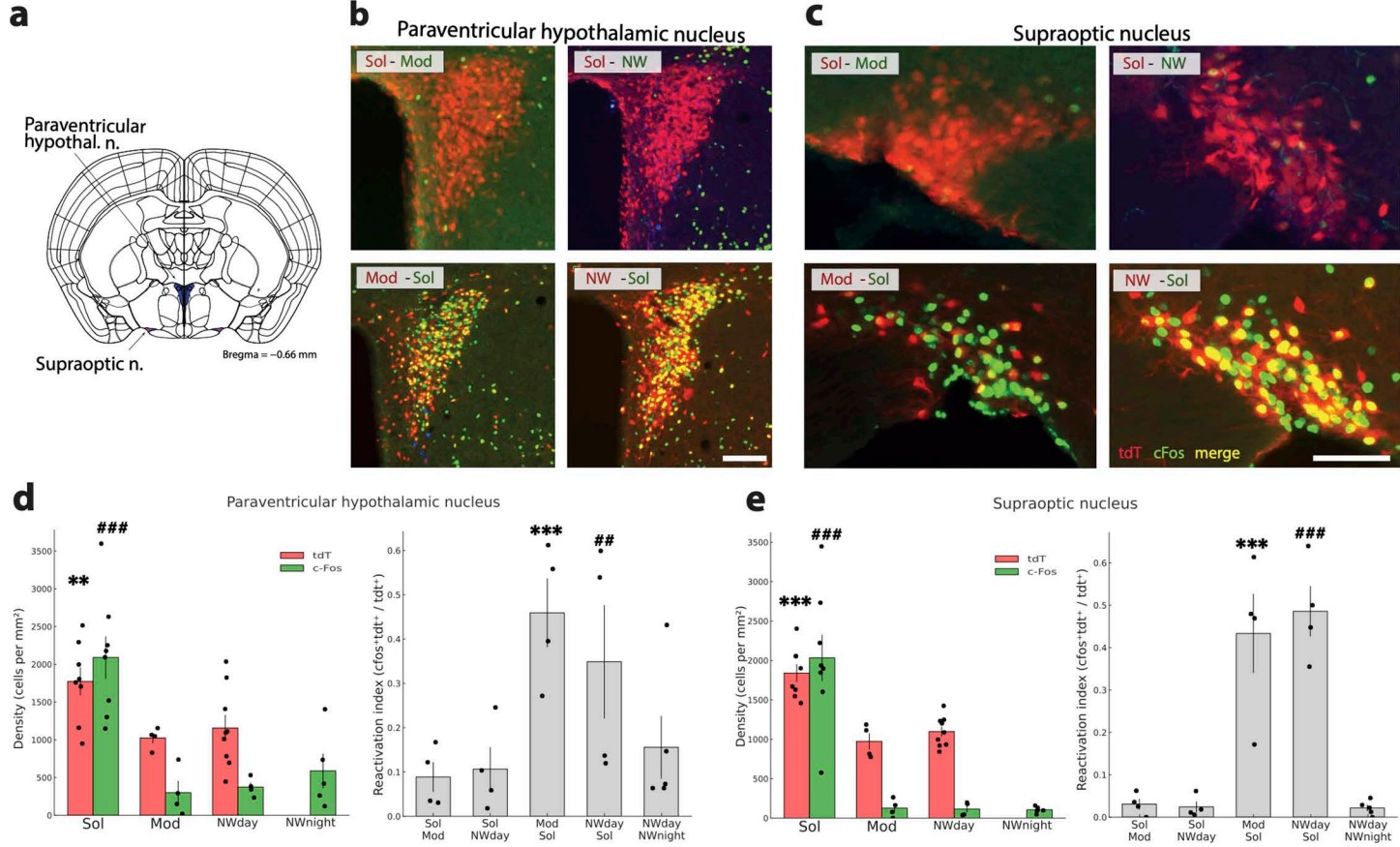

**Fig 8. Hypothalamic structures more activated by Solriamfetol. (a)** Drawings showing the localization of the two structures **(b, c)** Photomicrographs illustrating the distribution of tdT (red), c-Fos (green) and double-labeled neurons (yellow) in one representative mouse per experimental group in the paraventricular hypothalamic nucleus and supraoptic nucleus. **(d, e)** Histograms (left) showing the mean ± sem density (cell/mm²) of tdT+ (red) and cFos+(green) neurons with each mouse (dots) also displayed for the four conditions (Sol, Mod, Nwday, Nwnight). Note that for the paraventricular hypo-thalamic and the supraoptic nuclei, the density of cFos and tdT neurons is significantly higher in the Sol condition compared to the Mod, and NWday and NWnight conditions. Histograms on the right (gray) show the reactivation index (RI) in each group of mice (dots show each mice value). Note that the RI is significantly higher in the Mod-Sol and Nwday-Sol groups compared to their respective mirror groups. Significance tested with generalized linear models (Gamma family, log link) using robust (HC0) standard errors, one-sided in the "greater" direction with the animal as the experimental unit. Significantly different tdT density in the Sol condition vs. the other conditions: $p < 0.05$ * $p < 0.01$ ** $p < 0.001$ ***. Significantly different cFos density in the Sol condition vs. the other conditions: $p < 0.05$ #, $p < 0.01$ ##, $p < 0.001$ ###. Significantly different RI between Mod-Sol and Sol-Mod mice $p < 0.05$ * $p < 0.01$ ** $p < 0.001$ ***. Significantly different RI between Nwday-Sol and Sol-Nwday mice $p < 0.05$ #, $p < 0.01$ ##, $p < 0.001$ ###. Scale b,c: 100 μm; scale e: 60 μm. Raw data underlying the Figure is shown in S3 Data.

in Mod than in Sol and Nwday conditions, cFos density was higher in Mod than in Sol, Nwday, and Nwnight conditions; and the reactivation index (RI = cFostdt/tdt) was higher in Sol–Mod than in Mod–Sol groups. Applying these criteria, no structure met all requirements, indicating that none showed selective activation in Mod relative to the other conditions. A structure was classified as more activated in Nwday than in the other conditions if tdT neuron density was higher in Nwday than in Mod and Sol conditions, cFos density was higher in Nwday than in Nwnight, Mod, and Sol conditions and they showed a higher reactivation index (RI = cFostdt/tdt) in Sol–Nwday than in Nwday–Sol groups. Applying these criteria, no structure met all requirements, indicating that Nwday did not elicit selectively stronger activation than the other conditions. A structure was classified as more activated in Nwnight than in the other conditions if it did show a significantly higher cFos density in Nwnight than in Nwday, Mod, and Sol. Because tdT was not acquired in Nwnight, RI was not evaluated.

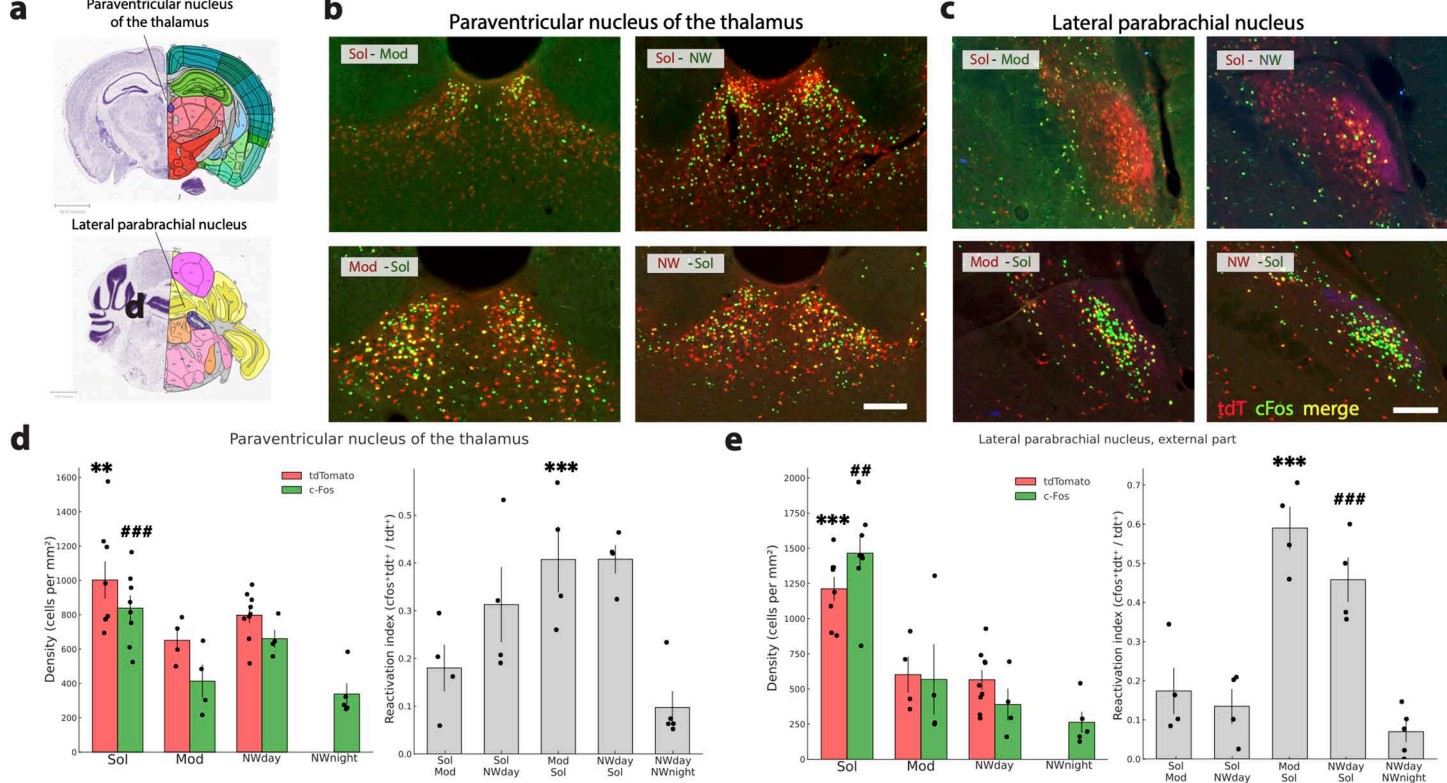

**Fig 9. Thalamic and pontine structures are more activated by Solriamfetol. (a)** Drawings showing the localization of the two structures **(b, c)** Photomicrographs illustrating the distribution of tdT (red), c-Fos (green) and double-labeled neurons (yellow) in one representative mouse per experimental group in the paraventricular nucleus of the thalamus and lateral parabrachial nucleus. **(d, e)** Histograms (left) showing the mean ± sem density (cell/mm²) of tdT+ (red) and cFos+(green) neurons with each mice (dots) also displayed for the four conditions (Sol, Mod, Nwday, Nwnight). Note that for the paraventricular nucleus of the thalamus and the external part of the lateral parabrachial nucleus, the density of cFos and tdT neurons is significantly higher in the Sol condition compared to the Mod, Nwday and Nwnight conditions. Histograms on the right (gray) show the reactivation index (RI) in each group of mice (dots show each mice value). Note that the RI is significantly higher in the Mod-Sol and Nwday-Sol groups compared to their respective mirror groups. Significance tested with generalized linear models (Gamma family, log link) using robust (HC0) standard errors, one-sided in the "greater" direction with the animal as the experimental unit. Significantly different tdT density in the Sol condition vs. the other conditions: $p < 0.05$ * $p < 0.01$ ** $p < 0.001$ ***. Significantly different cFos density in the Sol condition vs. the other conditions: $p < 0.05$ #, $p < 0.01$ ##, $p < 0.001$ ###. Significantly different RI between Mod-Sol and Sol-Mod mice $p < 0.05$ * $p < 0.01$ ** $p < 0.001$ ***. Significantly different RI between Nwday-Sol and Sol-Nwday mice $p < 0.05$ #, $p < 0.01$ ##, $p < 0.001$ ###. Raw data underlying the Figure is shown in S3 Data.

Under this definition, only the visual cortex showed a significantly higher density of cFos neurons in the NWnight compared to the three other conditions (not illustrated). A structure was classified as more activated in both Nwday and Nwnight relative to Sol and Mod conditions if cFos density was higher in both Nwday and Nwnight than in Mod and Sol conditions, tdT density was higher in Nwday than in Mod and Sol conditions and the reactivation index (RI = cFostdt/tdt) was higher in Sol–Nwday than in Nwday–Sol groups. Under these criteria, no structure showed a selective increased-activation in Nwday and Nwnight conditions relative to Mod and Sol conditions.

**Correlations between the densities of tdT and cFos-labeled neurons and locomotion.** For all structures, we determined whether there was a correlation between the density of tdT and cFos neurons and total locomotion. Among all structures, only four displayed a significant correlation between locomotion and neuronal density measured both with Tdt and cFos after Benjamini–Hochberg false-discovery-rate control ($q < 0.05$). These were the area postrema, lateral parabrachial nucleus, external part, nucleus of the solitary tract, caudal part, and supraoptic nucleus (Fig 10). Of great

Correlation with locomotion — Sol-specific (FDR-selected: q<0.05 & same-direction)

## Area postrema

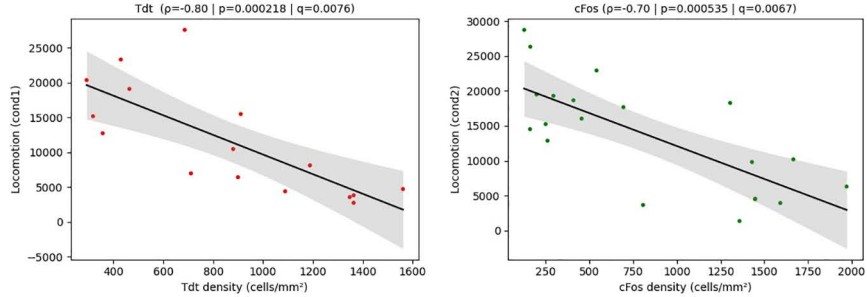

## Lateral parabrachial nucleus, external part

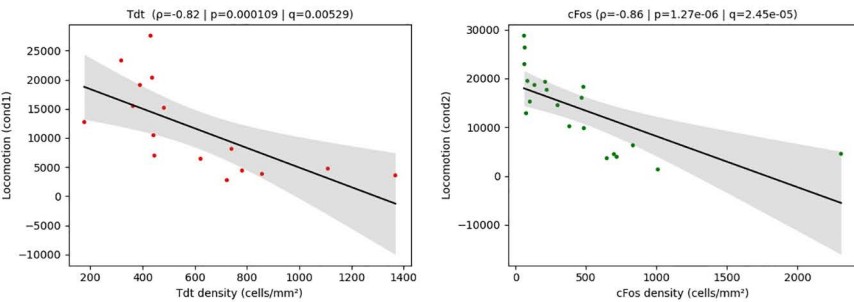

## Nucleus of the solitary tract, caudal part

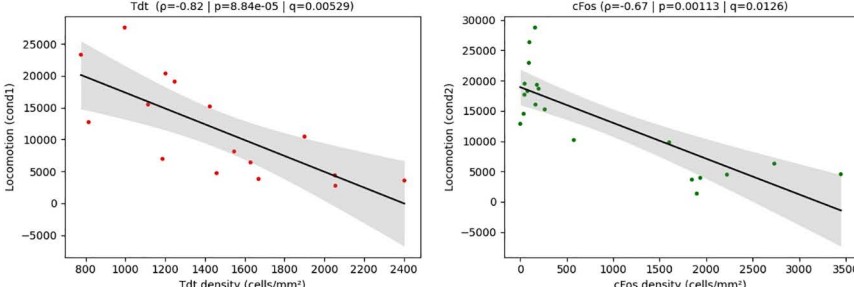

## Supraoptic nucleus

**Fig 10. Structures showing a correlation between densities of tdT and cFos and locomotion.** Panels illustrating the four Sol-specific regions that show a significant monotonic association between locomotion and neuronal density across both markers after Benjamini–Hochberg FDR ($q < 0.05$) with concordant signs. For each structure, left panel: tdT density vs. locomotion; right panel: cFos density vs. locomotion. Points are individual animals (tdT in

red; cFos in green). Black line: ordinary-least-squares fit; shaded band: 95% CI (for visualization only). Panel titles report Spearman's $\rho$, uncorrected $p$, and FDR-adjusted $q$. Densities were computed as counts/area (mm$^2$). Raw data underlying the Figure is shown in S1 and S3 Data.

interest, these four structures all showed a negative correlation with locomotion and did show an increased activation in the Sol condition.

The remaining five Sol-specific structures: bed nuclei of the stria terminalis, anterior division, oval nucleus, paraventricular hypothalamic nucleus, central amygdalar nucleus, lateral part, bed nuclei of the stria terminalis, and the paraventricular nucleus of the thalamus—did not pass the combined criterion ($q < 0.05$ for both Tdt and cFos). In these regions, non-selection arose from one or both of the following: (i) a lack of FDR-significance for one modality, and/or (ii) opposite correlation directions between Tdt and cFos. Thus, although these structures increased their activation during Sol, their inter-individual covariation between locomotion and neuronal density was not sufficiently robust or homogeneous across modalities to survive multiple-comparison control (S3 Fig).

**Differential activation of catecholaminergic and hypocretin/orexin neurons in Sol, Mod, or NW conditions.** To determine whether classical wake-promoting populations of neurons were differentially activated by Sol, Mod, and NW, triple labeling of tdT, cFos, and TH or orexin (Orx) was made. Then, the number of double and triple-labeled neurons was determined in all mice and groups.

For the catecholaminergic neurons, different levels of activation were observed among the different cell groups. To determine which TH-expressing nuclei were differentially activated between conditions, we quantified three complementary indices of neuronal activation: the proportion of TH+ neurons co-expressing tdT or cFos and the triple-labeling reactivation index measuring the probability that tdT-TH+ double-labeled neurons were reactivated (cFos+). Because triple labeling requires that both tdT and cFos derive from the same animal, only biologically valid condition pairs (Mod-Sol and Nwday-Sol) were included in the corresponding analyses, whereas Nwnight mice lacking tdT was retained only for descriptive comparisons in the cFos analysis. All statistical tests were performed using a binomial GLM applied directly to raw cell counts, followed by FDR correction (S3 Table).

Across the 11 structures examined, two nuclei exhibited a clear and consistent statistically significant increased number of TH+ neurons activated during Sol compared to all other conditions. In the lateral reticular nucleus (LRN, A1 NA group, Fig 11C and 11G), the proportion of tdT/TH neurons was significantly higher in Sol compared with both Mod and NWday, and the proportion of cFos/TH neurons was similarly increased relative to Mod, Nwday, and Nwnight. Critically, the triple-labeling analysis revealed that tdT-TH+ neurons in LRN were significantly more likely to reactivate during Sol than during either Mod or Nwday, indicating a strong functional bias toward Sol-associated activation (Fig 11G). Further, the percentage of TH+ neurons co-expressing tdT or cFos was over 55% in the Sol condition and below 30% in the Nwday and night and Mod conditions (Fig 11G and S3 Table).

A similar pattern was observed in the nucleus of the solitary tract (NTS containing the A2 NA group, Fig 11B and 11E). There was a significant increase in tdT-TH and cFos-TH labeling in the Sol condition compared with all other ones and the reactivation index demonstrated that tdT-TH+ neurons in NTS were robustly reactivated during Sol (cFos+) (Fig 11E). In addition, the percentage of TH+ neurons co-expressing tdT or cFos in all mice was around 30% in the Sol condition and below 20% or even lower in the other conditions. Together, these findings show that LRN and NTS form a pair of structures displaying a highly reproducible and Sol-specific recruitment profile.

The arcuate hypothalamic nucleus (Arc) also exhibited significantly more TH neurons activated (tdT-TH and cFos-TH) in the Sol than in the other conditions and the reactivation index demonstrated that tdT-TH+ neurons in NTS were robustly reactivated during Sol (cFos+). However, effect sizes were smaller and less consistent than those observed in LRN and NTS, suggesting that the arcuate nucleus displays a Sol bias but does not meet the full set of criteria defining the strongest Sol-recruited nuclei (S3 Table).

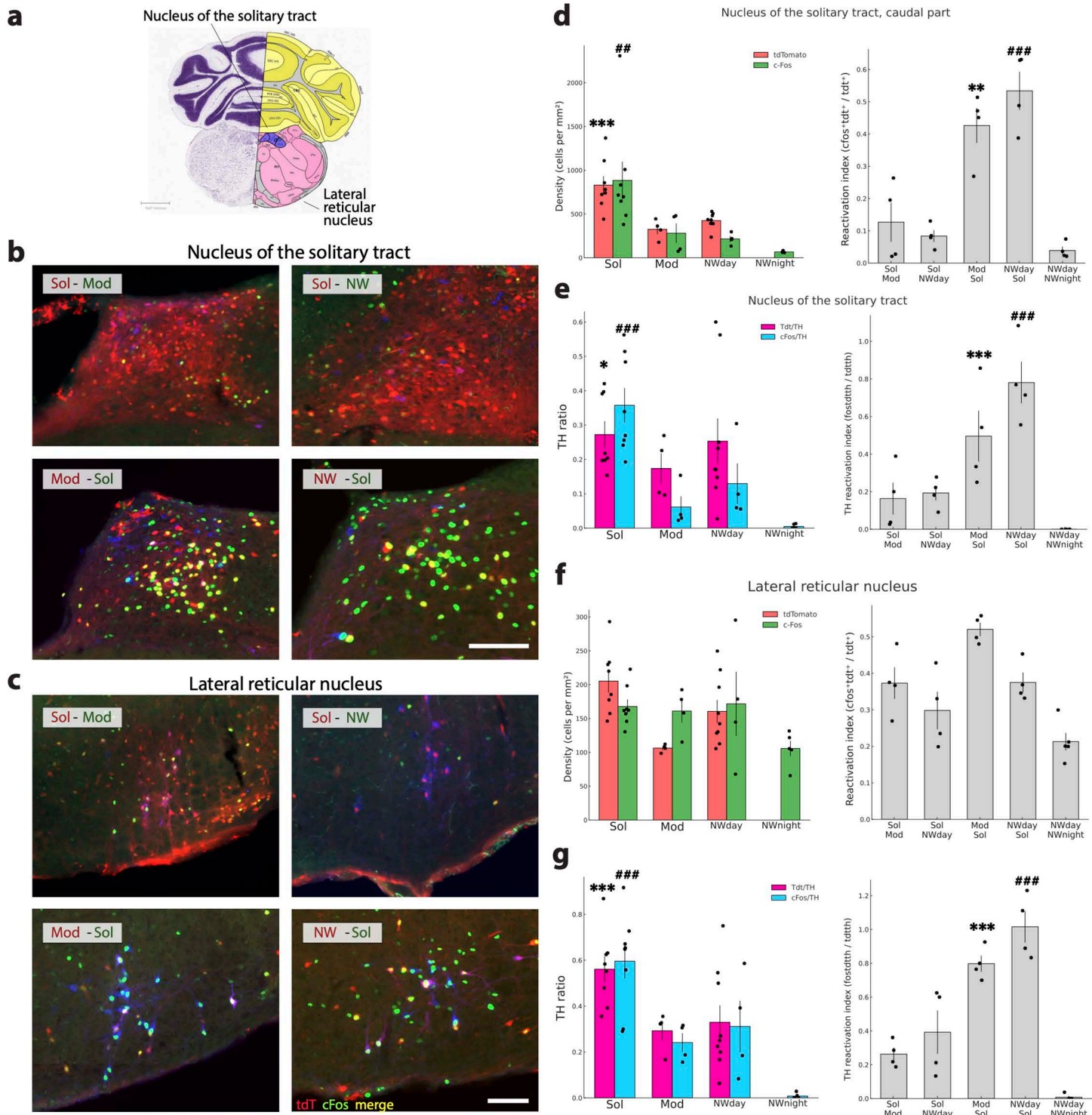

**Fig 11. Medullary noradrenergic structures more activated by Solriamfetol. (a)** Drawing showing the localization of the two structures **(b, c)** Photomicrographs illustrating the distribution of tdT (red), cFos (green), TH (blue), tdT and TH double-labeled neurons (purple) and, cFos and TH double labeled neurons (white nuclei) in one representative mouse per experimental group in the nucleus of the solitary tract and the lateral reticular nucleus. **(d, f)** Histograms (left) showing the mean ± sem density (cell/mm²) of tdT+ (red) and cFos+(green) neurons with each mouse (dots) also displayed for the four conditions (Sol, Mod, Nwday, Nwnight). Note that for the caudal part of the nucleus of the solitary tract but not for the lateral reticular nucleus,

the density of cFos and tdT neurons is significantly higher in the Sol condition compared to the Mod, Nwday and Nwnight conditions. Histograms on the right (gray) show the reactivation index (RI) in each group of mice (dots show each mice value). Note that only for the caudal part of the nucleus of the solitary tract, the RI is significantly higher in the Mod-Sol and Nwday-Sol groups compared to their respective mirror groups. **(e–g)** Histograms (left) showing the ratio of TH neurons double-labeled with tdT (pink) or cFos (blue) in the rostral part of the nucleus of the solitary tract (NTS) and lateral reticular nucleus (LRN). Note the significantly higher level of activation of the TH neurons in the Sol condition compared to the three other conditions. Gray histograms (right) showing the Reactivation index of TH neurons (tdT-cFos-TH+ neurons/ tdT-TH+ neurons) for LRN and NTS. Note that the RI is higher in Mod-Sol and Nwday-Sol conditions compared to their mirror conditions. Significance tested with generalized linear models (Gamma family, log link) using robust (HC0) standard errors, one-sided in the "greater" direction with the animal as the experimental unit. Significantly different tdT density or tdT-TH ratio in the Sol condition vs. the other conditions: $p < 0.05$ * $p < 0.01$ ** $p < 0.001$ ***. Significantly different cFos density or cFos-TH ratio in the Sol condition vs. the other conditions: $p < 0.05$ #, $p < 0.01$ ##, $p < 0.001$ ###. Significantly different RI between Mod-Sol and Sol-Mod mice $p < 0.05$ * $p < 0.01$ ** $p < 0.001$ ***. Significantly different RI between Nwday-Sol and Sol-Nwday mice $p < 0.05$ #, $p < 0.01$ ##, $p < 0.001$ ###. Raw data underlying the Figure is shown in S4 Data.

Other structures demonstrated only partial or inconsistent activation patterns. The locus coeruleus (LC) showed Sol-biased activation in TdT/TH and triple labeling, but not in the cFos analysis (Fig 12B, 12D, and 12E). The dorsal raphe, periventricular zone, and zona incerta each exhibited significant differences in one activation index but lacked cross-marker consistency (S3 Table). Finally, the ventral tegmental area, substantia nigra pars compacta, and area postrema showed no detectable Sol-related modulation (S3 Table). Further, the percentage of TH neurons activated was low in all these structures. The percentage of locus coeruleus TH+ neurons (LC, A6 noradrenergic, NA group, (Fig 12E) co-expressing tdT or cFos was indeed for most of the conditions around or even below 5%. In the dorsal raphe nucleus (DRN), the percentage of TH+ neurons co-expressing tdT or cFos was slightly higher than in the other dopaminergic structures (10%). The hypothalamic A11-A13 dopaminergic (DA) cell groups located in the zona incerta (ZI) consistently showed less than 3% of TH neurons co-expressing tdT or cFos across all conditions. Similarly, the ventral tegmental area (VTA, A10 dopaminergic (DA) group, Fig 12C and 12G) and the substantia nigra (SN, A9 DA group), both containing a very large number of TH-labeled neurons, showed very few double-labeled neurons. Indeed, less than 2% of the TH+ neurons in these nuclei expressed tdT or cFos, irrespective of the experimental condition.

Taken together, these analyses reveal that Sol induces a robust activation of TH neurons in the LRN and NTb, ehat is consistently observed across TdT, cFos, and reactivation measures. Arc displays a more moderate Sol enhancement, whereas all remaining structures show only partial or no evidence of Sol-specific activation.

We next examined whether locomotor activity was correlated with the activation of TH-activated neurons across brain structures. For most structures, no consistent relationship was observed between total locomotion and the activation of TH neurons. However, three structures exhibited a significant and convergent association between locomotor activity and TH neuron activation.

In the locus coeruleus, lateral reticular nucleus, and nucleus of the solitary tract, total locomotion was significantly negatively correlated with both TH–TdT and TH–cFos neuron counts. In all three regions, higher locomotor activity was associated with lower numbers of activated TH neurons, and this relationship was consistent across both labeling methods (Fig 13). These correlations remained significant after FDR correction and showed the same direction of effect for TH–TdT and TH–cFos, indicating a robust and convergent relationship between Sol treatment and TH neuron recruitment. Other structures either showed correlations for only one marker (TH–tdT or TH–cFos) or did not survive correction for multiple comparisons and were therefore not considered further. Together, these results identify a restricted set of three TH-containing brainstem nuclei in which locomotor activity is inversely related to TH neuron activation.

For the hypocretin/orexin (Hcrt-Orx) cell group located in the lateral hypothalamic area (LHA, (Fig 14 and S4 Table), the mean percentage of activated Hcrt-Orx neurons (i.e., labeled with tdT or cFos) across all mice was above 40% (Fig 13E), indicating a high level of activation during the Mod, Sol, and NWday and night conditions. Double-labeled Orx neurons constituted 36% and 21% of the tdT and cFos population in the LHA, respectively. To assess whether experimental conditions differentially engage Orx neurons in the lateral hypothalamic area, we combined double- and triple-labeling

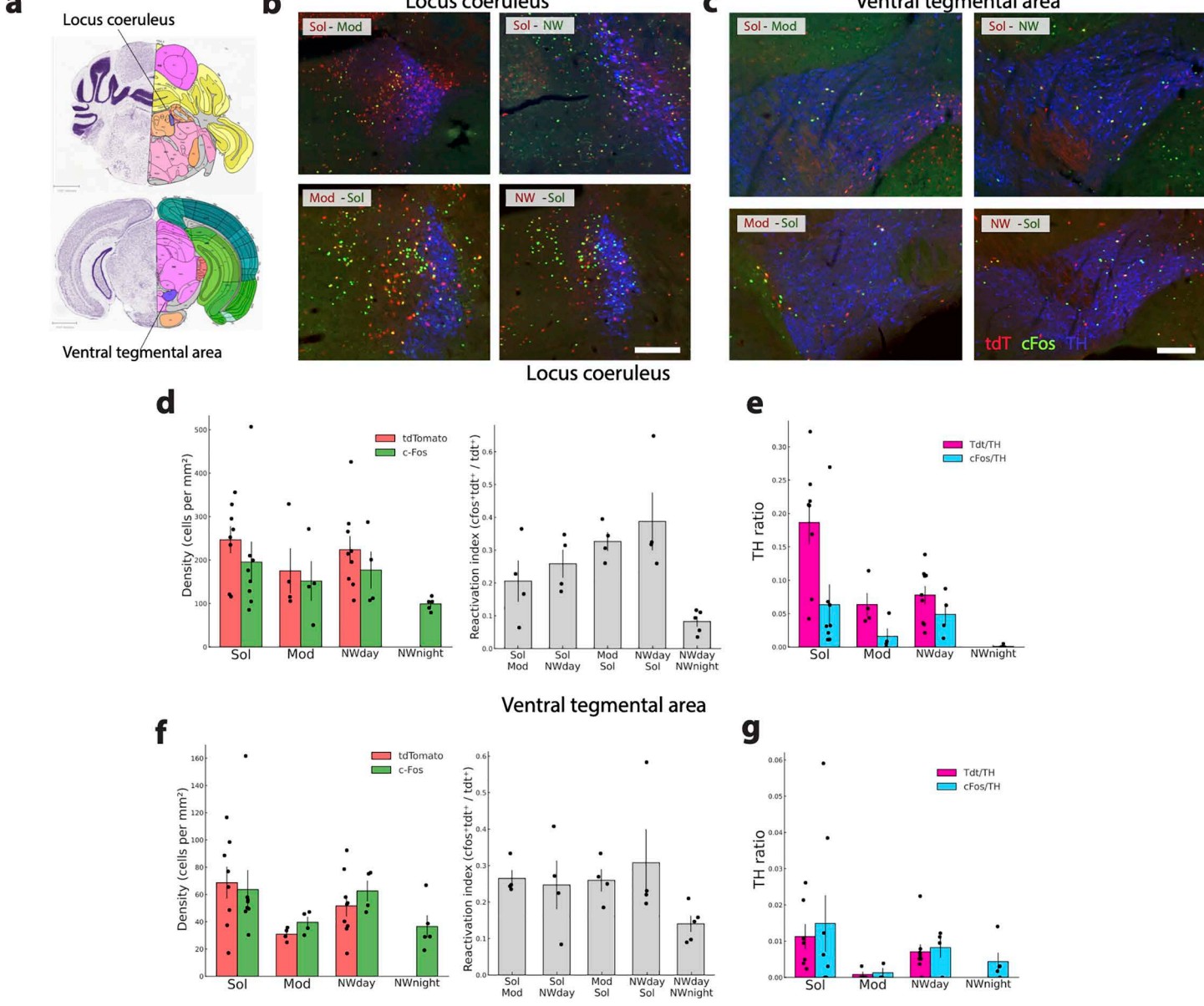

**Fig 12. Activation of the LC and VTA catecholaminergic groups during wakefulness. (a)** Drawings showing the localization of the two structures. **(b, c)** Photomicrographs illustrating the distribution of tdT (red), cFos (green), TH (blue), tdT and TH double-labeled neurons (purple) and cFos and TH double labeled neurons (white nuclei) in one representative mouse per experimental group in the LC and VTA. Note that in the LC, the number of TH and tdT or cFos double-labeled neurons is low in all conditions. A very low number of TH double-labeled neurons is also visible in the VTA in all conditions. **(d, f)** Histograms (left) showing the mean ± sem density (cell/mm²) of tdT+ (red) and cFos+(green) neurons with each mice (dots) also displayed for the four conditions (Sol, Mod, Nwday, Nwnight). Note that there is no statistical difference in the tdT and cFos densities in the LC and VTA across conditions. **(d, f)** The gray histograms show the reactivation index (tdTomato+/cFos+ neurons over total tdTomato+ neurons) for the two structures. There is no consistent difference across the four groups. **(e, g)** Histograms showing the ratio of TH neurons double-labeled with tdT (pink) or cFos (blue) in the LC and VTA. Note in all conditions the moderate activation in the LC and the very low level of activation of the TH neurons in the VTA. Significance tested with generalized linear models (Gamma family, log link) using robust (HC0) standard errors, one-sided in the "greater" direction with the animal as the experimental unit. Significantly different tdT density or tdTTH ratio in the Sol condition vs. the other conditions: $p < 0.05$ * $p < 0.01$ ** $p < 0.001$ ***. Significantly different cFos density or cFosTH ratio in the Sol condition vs. the other conditions: $p < 0.05$ #, $p < 0.01$ ##, $p < 0.001$ ###. Significantly different RI between Mod-Sol and Sol-Mod mice $p < 0.05$ * $p < 0.01$ ** $p < 0.001$ ***. Significantly different RI between Nwday-Sol and Sol-Nwday mice $p < 0.05$ #, $p < 0.01$ ##, $p < 0.001$ ###. Scale b: 300 μm; scale c: 200 μm. Raw data underlying the Figure is shown in S4 Data.

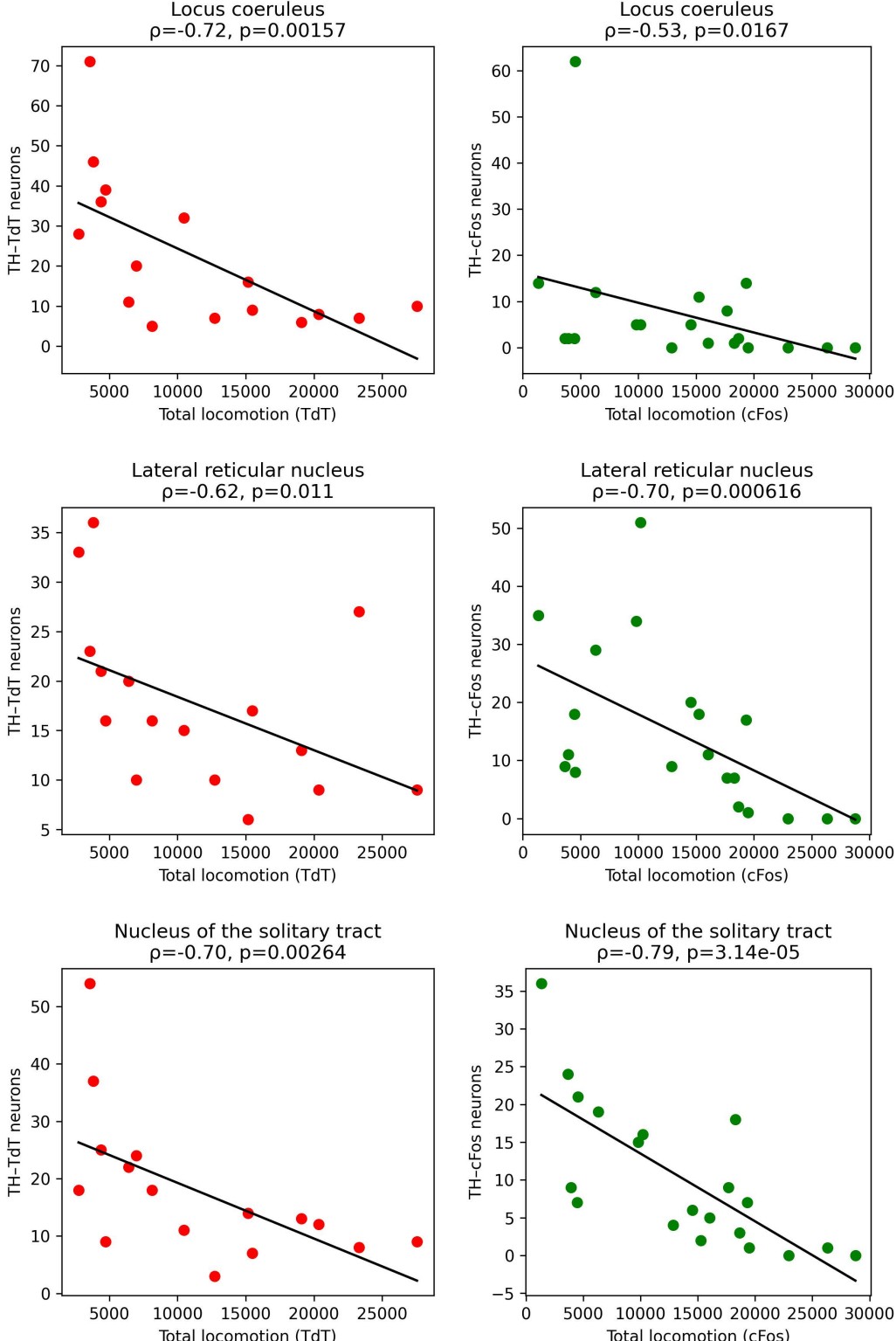

**Fig 13. Negative correlation between locomotor activity and activation of TH-positive neurons in three brainstem nuclei.** Scatter plots show the relationship between total locomotion and the number of TH-positive neurons labeled with TdTomato (TH–TdT, left column, red dots) or cFos (TH–cFos, right column, green dots) in the locus coeruleus, lateral reticular nucleus, and nucleus of the solitary tract. Each dot represents one animal. Black lines

indicate linear regression fits for visualization purposes. Spearman's rank correlation coefficients ($\rho$) and associated p-values are indicated in each panel. Only structures showing significant, FDR-corrected correlations for both TH–TdT and TH–cFos in the same direction are displayed. Raw data underlying the Figure is shown in S1 and S4 Data.

approaches and analyzed raw cell counts using binomial generalized linear models. Double-labeling analyses based on cFos expression revealed significant differences between conditions, with the proportion of cFos-positive Orx neurons being reduced in the Sol condition compared to Mod and Nwnight, while the comparison with Nwday did not reach significance (Fig 14E). Double labeling with tdT labeling displayed a similar direction of effect, with fewer tdT-positive Orx neurons in Sol compared to Nwday, although no significant difference was detected between Sol and Mod (Fig 14E). We next examined the reactivation of tdT-labeled Orx neurons using triple labeling (Fos + tdT + Orx among tdT + Orx neurons), focusing on mirror comparisons. In both Mod-Sol versus Sol-Mod and Nwday-Sol versus Sol-Nwday comparisons, reactivation probability was significantly lower when Sol occurred as the second episode (Fig 14E). This effect was consistent across both pairs, indicating an order-dependent reduction in Orx neuron reactivation associated with Sol. Taken together, these analyses indicate that the Sol condition tends to be associated with reduced activation of Orx neurons. However, this effect is not consistently observed across all labeling strategies and comparisons. We next examined whether locomotor activity was associated with the activation of orexin (Orx) neurons. No significant correlation was observed between total locomotion and the number of Orx neurons labeled with TdTomato (Orx–tdT). In contrast, total locomotion was positively correlated with the number of Orx neurons expressing cFos, indicating that animals with higher locomotor activity displayed increased recruitment of Orx neurons in the cFos condition (S4 Fig).

**Structures strongly activated in all conditions.** Our final aim was to identify populations of neurons strongly activated in the four different conditions namely Sol, Mod, and Nwday and night to identify candidate structures implicated in inducing wakefulness in all conditions. For that, we selected structures with an average RI superior to 0.3 or 0.2 in all of the four groups of mice (S5 Table). Structures with a density inferior to 50 cells/mm$^2$ in at least one mouse were discarded. Structures with a RI superior to 0.3 in the five groups of mice were the parasubthalamic hypothalamic nucleus (Fig 13C and 13F), the geniculate group-ventral thalamus implicated in the circadian system, and two nuclei from the sensory system, the inferior and the superior colliculus sensory related. Importantly, these structures had mean densities of cFos and tdT between 593 and 260 cells/mm$^2$. Then, 13 structures had a RI >0.2. Five hypothalamic nuclei were included in this group: the lateral hypothalamic area (Fig 13B and 13D), the posterior hypothalamic nucleus, the supramammillary nucleus and the ventral and dorsal premammillary nuclei. Additionally, seven brainstem nuclei namely the periaqueductal gray, the laterodorsal tegmental nucleus, the lateral reticular nucleus (Fig 11C and 11F), the rostral part of the nucleus of the solitary tract, the pretectal region and the superior colliculus, motor related and the tegmental reticular nucleus which are known to be involved in motor control were selected. Finally, the limbic basomedial amygdalar nucleus completed this group. In summary, our results reveal that an hypothalamic structure not previously implicated in wake control, namely the parasubthalamic nucleus, shows the highest level of activation across all conditions and therefore is a new strong candidate to drive wakefulness. In addition, we report additional potential players including some already identified like the lateral hypothalamic, the supramammillary nucleus, and the laterodorsal tegmental nucleus but also other unidentified structures like the posterior hypothalamic nucleus, the periaqueductal gray, the rostral part of the nucleus of the solitary tract, the lateral reticular nucleus and the ventral and dorsal premammillary nuclei. The other structures are known to be involved in more specific functions such as somatosensory regulation.

## Discussion

Our results indicate that selective subcortical structures are significantly more activated during wakefulness induced by Solriamfetol than by Modafinil or non-pharmacological natural day and night wakefulness suggesting that Sol induced a

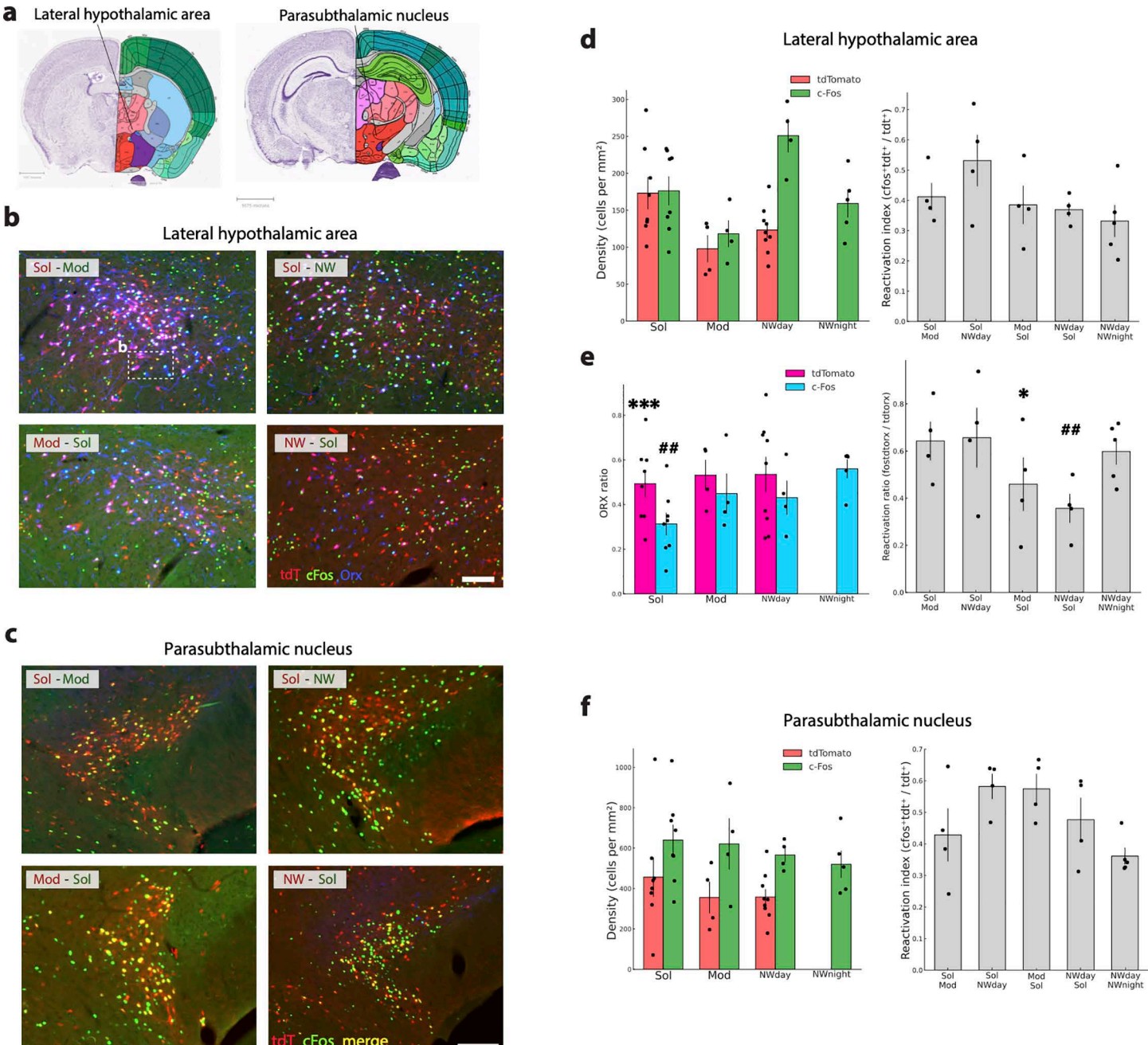

**Fig 14. The lateral hypothalamic area and the Hcrt/Orx neurons are strongly activated during wakefulness. (a)** Drawings showing the localization of the lateral hypothalamic area and the parasubthalamic nucleus. **(b)** Photomicrographs illustrating the distribution of tdT (red), cFos (green), Orx (blue), tdT and Orx double-labeled neurons (purple), and cFos and Orx double labeled neurons (white nuclei) in one representative mouse per experimental group. Note that in all groups of mice, many cFos, tdT and double-labeled and triple-labeled neurons are labeled in the LHA. **(c)** Photomicrographs illustrating the distribution in the PSTN of tdT (red), cFos (green) and double-labeled neurons (yellow) in one representative mouse per experimental group. Note that in all groups of mice, many cFos, tdT and double-labeled neurons are localized in the PSTN. **(d, f)** Histograms (left) showing in the LHA and PSTN the mean ± sem density (cell/mm²) of tdT+ (red) and cFos+(green) neurons with each mice (dots) also displayed for the four conditions (Sol, Mod, Nwday, Nwnight). **(d, f)** The gray histograms show the reactivation index (tdTomato+/cFos+ neurons over total tdTomato+ neurons) for the two structures. There is no consistent difference across the four groups of mice. **(e)** Histograms (left) showing the ratio of Orx neurons double-labeled with tdT (pink) or cFos (blue) in the lateral hypothalamic area. Note the significantly lower level of activation of the Orx neurons in the Sol condition compared to the other conditions. Gray histograms (right) showing the Reactivation index of Orx neurons (tdTcFos-Orx+ neurons/ tdT-Orx+ neurons). Note that

the RI is significantly lower in Mod-Sol and Nwday-Sol conditions compared to their mirror conditions. Significance tested with generalized linear models (Gamma family, log link) using robust (HC0) standard errors, one-sided in the "greater" direction with the animal as the experimental unit. Significantly different tdT density or tdTOrx ratio in the Sol condition vs. the other conditions: $p < 0.05$ * $p < 0.01$ ** $p < 0.001$ ***. Significantly different cFos density or cFosOrx ratio in the Sol condition vs. the other conditions: $p < 0.05$ #, $p < 0.01$ ##, $p < 0.001$ ###. Significantly different RI between Mod-Sol and Sol-Mod mice $p < 0.05$ * $p < 0.01$ ** $p < 0.001$ ***. Significantly different RI between Nwday-Sol and Sol-Nwday mice $p < 0.05$ #, $p < 0.01$ ##, $p < 0.001$ ###. Scale 100 μm. Raw data underlying the Figure is shown in S5 Data.

differential brain activation than the three other conditions. Solriamfetol also acutely decreased spontaneous ambulation and reduced theta-peak frequency. Correlational analysis revealed that increased locomotor suppression was associated with activation of a noradrenergic–hypothalamic network involved in arousal, interoception, and homeostatic regulation, rather than with engagement of classical mesencephalic locomotor or nigrostriatal motor circuits. In contrast, some subcortical structures were strongly activated during all four conditions suggesting some common mechanisms for wake induction in the four conditions. We discuss below the significance of these results in detail including methodological considerations and physiological implications.

## Methodological considerations

In the present study, we compared neuronal activation during waking induced by two drugs versus non-pharmacological means. To ensure that our control condition was corresponding to natural wake, we compared brain activation in Wday-Wnight mice. Indeed, wake is known to occupy most of the time during the early night in contrast to the day period during which it can be induced for only 2 hours using frequent sensory stimulation. We did not find a significant difference in neuronal activation in the NWday and Nwnight conditions across nearly all structures sampled indicating that the two conditions are closed to each other and represent natural wake. To our knowledge, our study is the first to use the TRAP mice to compare neuronal activation occurring during wakefulness induced by two different drugs versus natural means. More generally, our study is the first to extensively compare in the TRAP mice the distribution of the single and double-labeled neurons using the reporter gene (tdtomato) for the first condition and endogenous cFos expression for the second one. We first importantly showed that tdT labeling induced by the removal of the stop codon on the inserted tdTomato gene by the CreERT2 transgene under the cFos promoter after 4-OHT injection is comparable to natural cFos expression (See Fig 2). The obtention of tdT and cFos single and double-labeled cells in two different conditions in the same mice in two mirror groups (Mod-Sol versus Sol-Mod and NW-Sol versus Sol-NW) allowed us to strongly increase the reliability and the statistical power of our results. Indeed, structures were considered as modified by one condition only when (1) both tdT and cFos independent measurements and (2) the percentage of double-labeled neurons between the two mirror groups were significantly different. Using such an approach, we consistently found that a selected number of structures showed an increase in activation specifically in the Sol condition compared to the Mod, Nwday, and Nwnight conditions. Besides, another set of structures showed a high reactivation index in all groups of mice indicating that they are recruited during all types of wakefulness.

## Comparison with previous studies

In the present study, solriamfetol increased wakefulness like modafinil, but in contrast to modafinil and non-pharmacological wake, solriamfetol significantly reduced spontaneous ambulation without increasing delta power, while theta oscillations showed a marked reduction in frequency. It has been previously shown on the contrary that at high doses, modafinil (100 mg/kg) significantly increase locomotor activity while solriamfetol (150 mg/kg) has small effects [22]. However, it was also reported using similar high doses that solriamfetol induces an attentive wakefulness without pronounced locomotor activity [11]. These authors also reported in agreement with our results a reduction in theta peak frequency, reaching a minimum value (~5 Hz) 1–2 hours after drug administration, along with an increase in theta

power [11]. In the same study, they also compared the distribution of cFos induced by solriamfetol, modafinil, and non-pharmacological wakefulness. They reported that both drugs shared many areas of similar activation such as the lateral septum, the septohippocampal nucleus, the dorsal tenia tecta, the semilunar nucleus, the island of calleja, the olfactory tubercle, and the arcuate nucleus [11]. We also did not find differences in the activation of these structures between modafinil and solriamfetol. They further reported that CA1, the subiculum, ventral tenia tecta, the nucleus of solitary tract gelatinous, dorsal endopiriform, and suprachiasmatic nucleus were activated by solriamfetol, specifically. We did find significantly more neurons in the nucleus of the solitary tract, but not in the other structures. In contrast, a strong increase in the number of activated cells was seen in the oval part of the bed nucleus of the stria terminalis, the lateral part of the central amygdala, paraventricular hypothalamic and thalamic nuclei, supraoptic nucleus, external part of the lateral parabrachial nucleus, and area postrema. In summary, it is difficult to compare our results with the previous ones since the doses used were much higher (150 mg/kg versus 32 for Mod and 60 mg/kg for Sol) and the analysis was not detailed and lacked statistical analysis [11]. In addition, cFos labeling was analyzed in four previous publications after modafinil injection, but again the dosage was different than ours, the mapping was compared with saline injection (i.e., an undetermined state) and the analysis was not complete like in our experiments [23–25,26]. There were also a number of studies looking at cFos expression after periods of wakefulness induced by different methods in cats, rats, and mice, but the mapping was either not complete or not detailed enough to compare the data with the present study [27–29]. Finally, our study is the first in which a direct comparison is made in the same mouse between drugs and non-pharmacological wakefulness.

## Physiological significance

Our results show that the bed nucleus of the stria terminalis (BNST) particularly its oval part, lateral part of the central amygdalar nucleus (CEA), paraventricular thalamic (PVT) and hypothalamic nuclei (PVN), supraoptic nucleus (SON), external part of the lateral parabrachial nucleus (LPB), caudal part of the nucleus of the solitary tract (NTS) and area postrema are significantly more activated by solriamfetol than modafinil and non-pharmacological wakefulness. Further, the number of tdT and cFos neurons labeled in these structures was inversely correlated with the distance traveled. Such results are both new and original since it is traditionally thought that wake is induced by classical structures such as the hypocretin/orexin and histaminergic neurons, dorsal raphe serotonergic neurons, telencephalic and pontine cholinergic neurons and the noradrenergic neurons of the locus coeruleus [4]. However, most of the structures increased by solriamfetol reported in the present study have been shown to play a role in wake mostly in very recent publications (see below). Further, there is ample evidence that these structures are interconnected (see below). Indeed, optogenetic or chemogenetic excitation of GABAergic neurons of the BNST induces wake and such effect is attenuated by pretreatment with orexin antagonist [30]. The authors showed that the BNST strongly projects to other structures highly activated in the present paper such as the CEA, PVN, SON, LHA, NTS, and LPB.

Using cFos and unit recordings, it has been previously shown that PVT neurons are more active during wake than sleep (REM and moreover NREM sleep, [7]. Optogenetic activation of PVT neurons and their projections to the nucleus accumbens induces wake whereas their lesion and optogenetic or chemogenetic inactivation reduces it, while increasing NREM during the dark period [7]. Also, chemogenetic inactivation of the hypocretin/orexin (Orx) projection to the PVT decreases wake and increases NREM. Nevertheless, wake induced by stimulating Orx neurons was not abolished by PVT chemogenetic inactivation, indicating that other pathways are involved [7]. On the other hand, calcium imaging showed that PVT neurons projecting to the CEA are specifically activated during wake and that their optogenetic activation induces wakefulness while their inhibition increases NREM sleep [31]. Such pathway may mediate stress response during wake, since their optogenetic inhibition also alleviated the hormonal and behavioral responses to acute stress [31].

The paraventricular hypothalamic nucleus (PVN) which is strongly activated by solriamfetol was previously shown using cFos or calcium imaging to contain glutamatergic (including vasopressin, oxytocin, and corticotropin-releasing factor (CRH) subtypes) neurons active during wake and inactive during sleep [5]. Optogenetic activation of these neurons or

their projections to the lateral parabrachial nucleus (LPB) and lateral septum (LS) induced wake while their chemogenetic inhibition or ablation reduced wake and increased NREM [5]. It has also been shown that chemo and optogenetic activation specifically of the vasopressin subtype of PVN neurons induces wake while their inhibition decreases it. The increase in wake was also obtained by stimulating PVN terminals in the lateral hypothalamic area and was strongly decreased by pretreatment with an Orx antagonist [32]. It was also shown that restraint stress causes strong cFos expression in corticotropin-releasing hormone neurons (CRH) of the PVN and also of Orx neurons [33]. Optogenetic stimulation of the CRH neurons of the PVN projecting to the lateral hypothalamic area leads to insomnia while inhibition of the pathway compromises restraint stress-induced insomnia [33]. Besides, chemogenetic and optogenetic activation of glutamatergic and calcitonin gene-related peptide (CGRP) neurons of the external lateral parabrachial nucleus was also shown to induce W [34]. Conversely, optogenetic inactivation of these neurons or of their projections to the BNST, CEA, and LHA increased the latency to arousal induced by hypercapnia [34]. In addition, our study identified additional structures never implicated in wake control before such as the supraoptic nucleus (SON), caudal part of the nucleus of the solitary tract (NTS), and area postrema. Interestingly, these structures have known direct neuroanatomical links to those mentioned above and therefore seem to be part of the network strongly activated by Sol.

In summary, recent published data strongly fit with our results indicating that the structures activated by Sol revealed in the present study form a network likely inducing wake (Fig 15). Importantly, these structures are significantly less activated during NW, and moreover, Mod indicating that Sol has a different mode of action than Mod. This is surprising since both drugs are known to be norepinephrine and dopamine reuptake blockers. However, it has recently been shown that Sol but not Mod has additional agonist activity at the trace amine associated receptor 1 (TAAR1) [13]. Besides, it has been shown that a TAAR1 agonist induces W and inhibits NREM sleep [14]. In addition, increased cFos staining has been reported after injection of two different TAAR1 agonists in several structures specifically activated by Sol in our study namely the bed nucleus of the stria terminalis, central amygdala, lateral parabrachial nucleus, nucleus of the solitary tract, area postrema, parasubthalamic nucleus (PSTN) and lateral hypothalamic area [35]. Altogether, these data strongly suggest that part of the wake-inducing effect of Sol is mediated through activation of a set of structures by its TAAR1 receptors agonist activity.

In addition, the parasubthalamic nucleus, lateral hypothalamic area, laterodorsal tegmental nucleus, posterior hypothalamic nucleus, supramammillary nucleus, the ventral and dorsal premammillary nuclei, the limbic basomedial amygdalar nucleus, geniculate group of the ventral thalamus, superior colliculus, sensory and motor related, the pretectal region, tegmental reticular nucleus, the periaqueductal gray, the rostral part of the nucleus of the solitary tract, and the lateral reticular nucleus showed a high reactivation index in all groups of mice indicating that they are strongly activated during wake induced by Sol, Mod and NW. The geniculate group of the ventral thalamus and the inferior and superior colliculus, sensory-related nuclei are known to be implicated in circadian or sensory processing and are therefore unlikely involved in specifically inducing wake. In contrast, some of the other structures identified have been recently implicated in wake induction. Indeed, glutamatergic neurons of the parasubthalamic nucleus (PSTN) are active during W and REM sleep but not NREM [6]. The optogenetic and chemogenetic activation of these neurons and their projection to the VTA and lateral parabrachial nucleus increases wakefulness and exploratory behavior while their inhibition decreases wakefulness [6]. The PSTN receives an excitatory projection from parabrachial glutamatergic neurons. The PSTN also projects to the bed nucleus of the stria terminalis, central amygdala, and PVT [6]. The lateral hypothalamic area (LHA) also contained a large number of neurons activated by Sol, Mod, and NW. We further showed in the present study that 26% of these neurons were Hcrt/Orx neurons. The role of Hcrt/Orx neurons in wake induction and maintenance is supported by a large number of studies showing using cFos and unit recordings that they are specifically active during W and that their chemogenetic and optogenetic activation induces W, while their inhibition or ablation leads to hypersomnia [4]. In addition to the Hcrt/Orx neurons, a population of GABAergic neurons of the LHA have been shown to be W active and their optogenetic and chemogenetic activation as well as that of their projection to

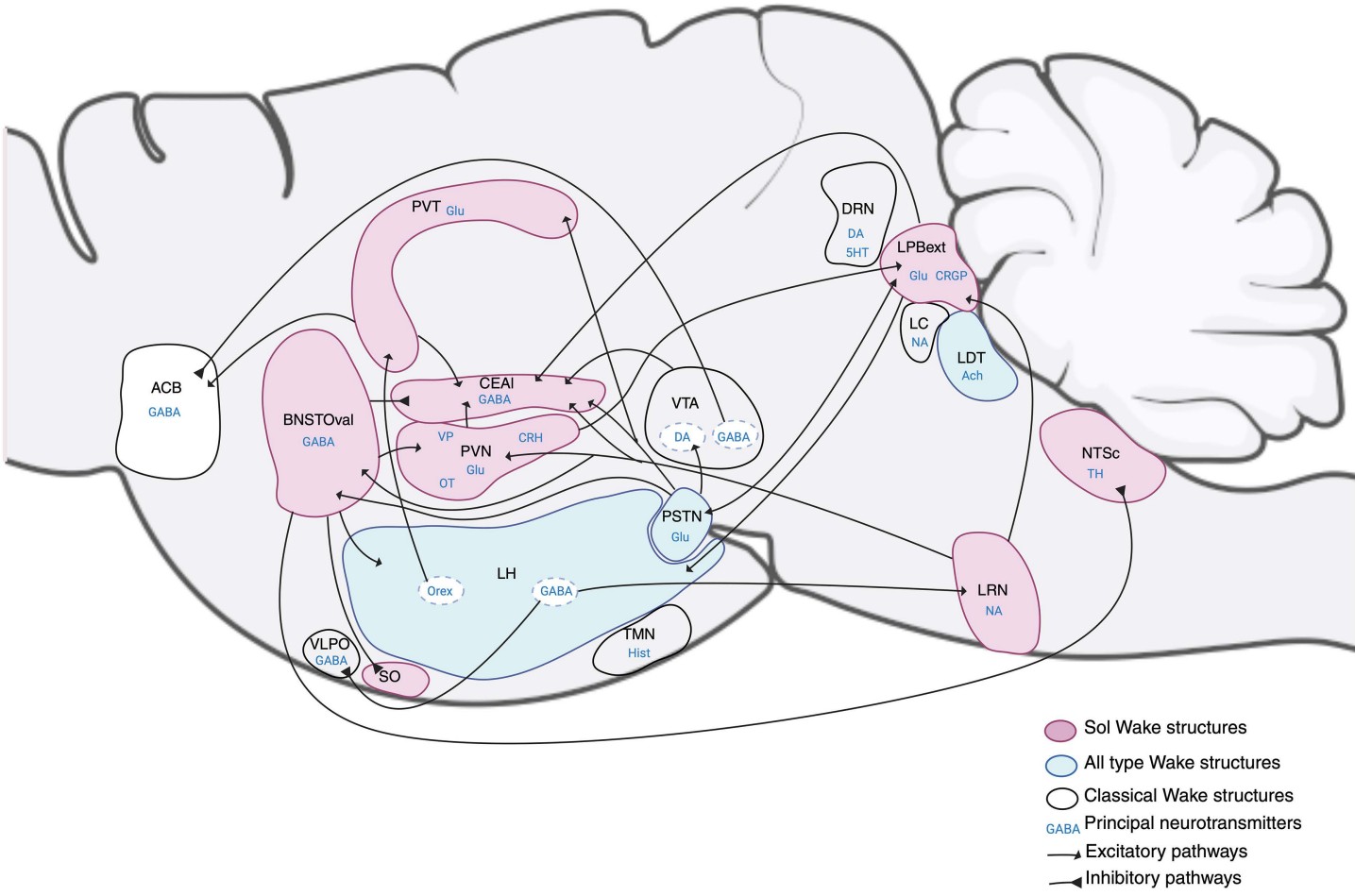

**Fig 15. The neuronal network of the wakefulness system activated by Sol, Mod and NW.** Drawing showing many interconnected structures implicated in wakefulness, after Sol, and Mod injections and NW. The pink structures are significantly more activated by Sol than Mod and NW. Sol strongly activates a subset of structures less activated in non-pharmacological wakefulness (NW) and only weekly activated by Mod. The blue structures are activated by Sol, Mod and NW. The white ones are known to be implicated in the wake system, for the DRN, LC and TMN, their role is minor based on our results. Different neurotransmitters are involved in this system inhibiting or exciting wake structures, Like gamma-aminobutyric (GABA), glutamate (Glu), norepinephrine (NA), Histamine (Hist), acetylcholine (Ach), Orexin (Orx), dopamine (DA), serotonin (5HT), oxytocin (OT), vasopressin (VP), corticotropin-releasing hormone (CRH), and calcitonin gene-related peptide (CRGP). ACB: accumbens nucleus, BNSTOval: bed nucleus of the stria terminalis, anterior division, oval nucleus, CEAI: central amygdalar nucleus, lateral part, DRN: dorsal raphe nucleus, LC: locus coeruleus, LDT: laterodorsal tegmental nucleus, LH: lateral hypothalamus, LPBExt: lateral parabrachial nucleus, external part, LRN: lateral reticular nucleus, NTSc: nucleus of the solitary tract, caudal part, PSTN: parasubthalamic nucleus, PVN: paraventricular nucleus of the hypothalamus, PVT: paraventricular nucleus of the thalamus, SO: supraoptic nucleus, TMN: tuberomammillary nucleus, VLPO: ventrolateral preoptic area, VTA: ventral tegmental area. Created in BioRender. Team, S. (2026) https://BioRender.com/0ltxrc4.

sleep-active structures such as the lateral reticular thalamic nucleus and the ventrolateral preoptic area induced wake [36,37]. Finally, it has also been previously shown that the laterodorsal tegmental nucleus, in particular its cholinergic neurons, plays a role in W induction [38]. The chemogenetic activation and inhibition of the supramammillary nucleus was also shown to increase or decrease wakefulness, respectively [39]. Finally, to our knowledge the posterior hypothalamic nucleus the ventral and dorsal premammillary nuclei, the periaqueductal gray, the rostral part of the nucleus of the solitary tract, and the lateral reticular nucleus have not been previously involved in W and additional data is needed to determine whether they could play a role.

We did also determine whether the dopaminergic and noradrenergic systems are activated during wake induced by Mod, Sol, and NW. Surprisingly, no more than 4% of the dopaminergic neurons of the hypothalamic A11 and A13 (zona incerta) and the mesencephalic A10 (VTA) and A9 (substantia nigra) groups were activated in all conditions. These results are in line with our previous cFos study in rats showing that these neurons are not activated during W [40]. They are also supported by calcium imaging of DA neurons of the VTA showing their low activity during W although it was higher than during NREM sleep [41]. Nevertheless, optogenetic activation of these neurons or their projection to the nucleus accumbens, the striatum, and the central amygdala induced W while their inhibition promoted nest building, induced a decrease in wake and increase in NREM indicating that despite their low activity, they play a role in wake induction [41]. In the dorsal raphe nucleus (DRN), less than 12% of the dopaminergic neurons did express tdT or cFos in all three wake conditions suggesting that they might play a minor role in inducing the wake state. Fiber photometry reported that dopaminergic neurons of the DRN are indeed specifically active at the onset of W. Optogenetic activation of dopaminergic neurons of the DRN induced wake while their chemogenetic inactivation decreased it and increased NREM [42]. Based on these and our data, it is likely that only a subset of DA neurons of the DRN are involved in wake control.

In the case of the LC noradrenergic neurons, NA A6 group), less than 12% of the neurons were activated in all conditions. In our previous study in rats, we reported that 30% of the noradrenergic LC neurons were cFos+ after wake induced by a novel environment with two rats in the arena [43]. Further, we found for the first time that LC NA activation was inversely correlated with locomotor activity. The higher number of double-labeled NA LC neurons in the previous study might be due to the stronger stimulation induced by social exposure. Optogenetic excitation of LC noradrenergic neurons induced wake while their inhibition induced a small reduction of wake [44]. Further, optogenetic inhibition decreases sensory evoked awakenings [45]. Interestingly, cFos staining induced during wake was strongly decreased in the cortex but not the hypothalamus after LC lesion [46]. Altogether, these results suggest that LC noradrenergic neurons directly activate the cortex during wake but might play a more marginal role in inducing wake by interacting with the other subcortical wake-inducing systems. In summary, our data indicate that classical wake-promoting systems such as the LC noradrenergic and the VTA dopaminergic neurons are not strongly activated during long-lasting wakefulness induced by modafinil, solriamfetol, and sensory stimulation. It might be hypothesized that they are implicated in inducing arousal after acute stimuli and not long-lasting wake although additional experiments are needed to confirm such hypothesis.

In the nucleus of the solitary tract (A2 NA group), the mean percentage of TH neurons co-expressing tdT or cFos was 30% for the Sol condition, 10% for the Mod condition, and 18% for the NW condition. In the lateral reticular nucleus (A1 NA group), the median percentage of TH neurons co-expressing tdT or cFos was superior to 50% in the Sol condition and between 20% and 40% in the NW and Mod conditions. In our previous study in rats, around 20% of the noradrenergic neurons of A1 and A2 groups were labeled with cFos after wake [43]. Although the role of A1 and A2 in inducing W is mentioned in a review on noradrenergic functions [47], no study is devoted to such involvement. Interestingly, A1 and A2 noradrenergic neurons are strongly projecting to the BNST, CEA, PVN, SO, and LPB [48] shown here to be statistically more activated by Sol than NW or Mod. Finally, we show for the first time that the activation of A1 and A2 noradrenergic groups is inversely correlated with locomotion. From these results, we propose that A1 and A2 noradrenergic projections are playing a key role in inducing wake associated with a decrease in locomotion in the Sol condition while they play a more modest role in wake induction by Mod or in the Nwday and moreover the Nwnight condition.

## Conclusions

In summary, our results disclose for the first time over the entire brain the neuronal systems activated by Sol, Mod, and NW. In particular, we showed that Sol strongly activates a subset of structures not strongly activated in NW and Mod. These results indicate that W induced by Sol might be different from that induced by Mod and NW. It suggests that patients treated by Sol might be in a different neuronal state than those treated by Mod. Further, our results indicate that a large number of interconnected structures not classically involved in inducing W might play a major role in inducing

the state. We created a schematic drawing showing the structures identified in the present study and the pathways likely involved ([Fig 15]). Our results therefore introduce a new concept that wake induction might be due to many more structures than previously thought and is not only induced by classical catecholaminergic, serotonergic, histaminergic, and hypocretin systems. It opens the way to future studies to determine the respective roles of the newly discovered structures in wake and to study the specificity of the wake state induced by Sol.

## Materials and methods

### Animals

All experiments were conducted in accordance with the French and European Community guidelines for using animals in research and were approved by the institutional animal care and committee of the University of Lyon 1 and NEURO-CAMPUS (APAFIS #31722-2021051809304729). Both male and female of double heterozygous Fos2A-iCreER;R26Ai14 (TRAP2) mice, kindly gifted by Dr. Liqun Luo from Stanford University were used. TRAP2-RED mice were generated by crossing Fos2A-iCreER/+ (TRAP2) mice to R26AI14/+ (AI14) mice [1]. In all experiments, 8–12 weeks old mice were prepared for surgery. Transgenic mice were housed individually and placed under a constant light/dark cycle (light on from 7:00 am to 7:00 pm) and were habituated to be handled every day.

### Surgery

Mice were anesthetized with Ketamine and Xylazine (100/10 mg/kg, i.p.). Then, the top of the head was shaved, and the mice were placed in a stereotactic frame with a heating pad underneath. Two stainless screws were fixed in the parietal part (AP: −2.0 mm, ML: 1.5 mm from bregma) and one in the frontal part (AP: +2.0 mm, ML: 1.0 mm from bregma) of the skull, whereas the reference electrode for unipolar EEG recording was fixed in the occipital part (AP: −5 mm, ML: 0.0 mm from bregma). Two wire electrodes were inserted into the neck muscles for bipolar electromyogram (EMG) recordings. All leads were connected to a miniature plug (Plastics One, Roanoke, VA) that was cemented on the skull.

### Polysomnography and behavioral recordings

Animals were allowed to recover from surgery for 5 days in their home cage before being habituated to the recording conditions for seven days. They were then connected to a cable attached to a slip-ring commutator to allow free movement within the recording barrel. Unipolar EEG and bipolar EMG signals were amplified, respectively, 1:5,000 V/V and 1:2,000 V/V (MCP-PLUS, Alpha-Omega Engineering, Israel), digitized at 1,024 Hz, and acquired using Slip Analysis v 2.9.8 software (Viewpoint, Civrieux, France). Behavioral recordings were made using a digital camera placed above the recording barrel and stored for further analysis (see below). Baseline recordings were made during 48 h starting just after the 3-day habituation period. During these baseline recordings, general behavior and the sleep-wake cycle were monitored to guarantee free and natural behavior inside the recording barrel. At 10 am of the first experimental day (first session), animals were treated with solriamfetol, modafinil, or underwent sensory stimulation (either light touch with a upon immobility or placement of novel objects in the cage). After two hours, all mice received 4-hydroxytamoxifen (4-OHT) i.p. injection (circa 12 pm). Seven to 10 days later, the second experimental section started around 10 AM with the pharmacological or non-pharmacological waking. Animals were perfused 2 hours later (circa 12 pm). In the day-night wake paradigm, animals were non-pharmacologically stimulated at 10 am (Nwday, first session) and 10 pm (Nwnight, second session), receiving 4-OHT or perfused, respectively, two hours later.

### Behavioral and electrophysiological analysis

To evaluate the wake-promoting effects of solriamfetol and modafinil and compare them with natural waking, mice underwent two-hour simultaneous video-electroencephalography (EEG) and electromyography (EMG) recordings following

each pharmacological treatment. These conditions were compared with non-pharmacological wakefulness (NW) induced by gentle sensory stimulation. Experiments were conducted in two sessions, the first revealed by tdT staining and the second, by cFos (Fig 1). Additional mice were used to investigate whether wake induction in the light and dark periods had a differential effect on cell density in animals experiencing non-pharmacological wakefulness (S1 Fig).

Behavioral and electrophysiological analysis were made using custom-made algorithms in Matlab (Mathworks, USA). Spontaneous locomotor activity was quantified throughout the two-hour recording session using computer-assisted video tracking (MouseLabTrack, [49]). The animal's position was used to compute the occupancy heat maps and locomotor trajectories. Locomotion velocity and immobility epochs were extracted from position data, and total distance traveled was calculated for each animal across the entire session. EEG and EMG signals were recorded continuously during the same two-hour period. Vigilance states in the 2 hours after experimental treatment were visually scored in 5 s episodes as previously described [50] and the amount of wakefulness, slow wave sleep (SWS) and REM sleep were computed. Time–frequency spectrograms (0–20 Hz, *spectrogram* function) were computed to characterize oscillatory dynamics during wakefulness. Power spectral density (PSD, *pwelch* function) was averaged across the full recording and within three predefined 10-minute epochs: immediately after treatment (0–10 min), mid-session (55–65 min), and late session (110–120 min). For each epoch, delta (slow-wave, 0.5–4 Hz) and theta-band (5–10 Hz) activity were quantified by calculating band-specific power and peak frequency. These measures were used to assess whether differences in locomotor activity were associated with changes in arousal level or sleep intrusion.

## TRAPing

4-hydroxytamoxifen (4-OHT) was prepared as described previously [19]. Briefly, 4-OHT (Cat# H6278 Sigma Aldrich, St. Luis, MO) was dissolved at 20 mg/mL in absolute ethanol by ultrasonic water bath at 37 °C for 10 min and was then aliquoted and stored at −20 °C as a stock solution. Before injection, corn oil (Sigma Aldrich) was added to the thawed stock solution to replace the ethanol and to obtain 10 mg/mL 4-OHT, and then ethanol was evaporated at 37 °C. The 10 mg/mL 4-OHT solution was used the same day or one day after preparation. All animals were injected intraperitoneally (i.p.) with 80 mg/kg 4-OHT, 2 hours after the beginning of the first Wakefulness (W) induction.

## Methods for inducing wakefulness

Solriamfetol was dissolved in saline solution (NaCl 0.9%). Modafinil was dissolved in a vehicle containing dimethyl sulfoxide (DMSO) 3% and sterile saline (NaCl 0.9%) and was then injected immediately to each mouse. Drug solutions were freshly prepared the day of the injection. Solriamfetol and modafinil were administered by intraperitoneal injection at a concentration of 10 ml/kg body weight at 10 am. Wake onset was defined as the time elapsed between the time of injection and the first wake episode lasting at least 1 min and not interrupted by more than two 4 s epochs scored as NREM sleep. We did use 60 mg/kg for Solriamfetol (Sol) and 32 mg/kg for Modafinil (Mod) to obtain an induction of wakefulness (W) lasting around two hours. By comparison, Hasan and colleagues [11] used 150 mg/kg for both Modafinil and Solriamfetol which are high doses possibly inducing non-specific side effects and jeopardizing conclusions regarding cFos activation. The NW protocol was used to induce wakefulness during 2 hours (before perfusion) or 4 hours for 4-OHT. In this case, 4-OHT was injected after two hours of wake and the mice were left in the open field two more hours to avoid TRAPing of neurons activated during sleep before the washout of the drug. When the mice were perfused, they were left in the open field for two hours before perfusion. To maintain the animals awake, we did put in their barrels wood tips and small objects. Food and water were freely available in the barrels. During non-pharmacological wake, the animals were permanently monitored by a web-camera from a different room to check whether they were staying awake. The animals were gently touched by a soft tissue when they became inactive/drowsy. Five groups of mice were generated. In the first group, solriamfetol was injected at 10a.m and 4-OHT was injected two hours after the beginning of the induction of wake. One week later, modafinil was injected and the

mice were perfused after two hours of wakefulness (Sol-Mod group). In the second group of mice, the two protocols of waking induction were inverted (modafinil first and solriamfetol second, Mod-Sol group). In the third group, the animals were left in their barrel and stimulated when they were starting to be drowsy and 4-OHT was injected two hours after the beginning of the induction of wakefulness. One week later, solriamfetol was injected at 10 am and the mice was perfused after at 12 am two hours of wakefulness (NW-Sol group). In the fourth group of mice, the two protocols of wake induction were inverted (solriamfetol first and non-pharmacologic wake second, Sol-NW group). The mice were perfused at 12 am 2 hours after the beginning of the NW protocol. In the fifth group, the animals were left in their barrel and stimulated when they were starting to be drowsy and 4-OHT was injected at 12 am two hours after the beginning of the induction of wakefulness. One week later, non-pharmacological wakefulness protocol started at 7 pm and the mice were perfused after two hours of wakefulness (Nwday-Nwnight group).

## Perfusion

All animals were deeply anesthetized with intraperitoneal (IP) injection of pentobarbital (140 mg/kg) and were perfused with a heparin-added Ringer's lactate solution (1:1,000) followed by 4% paraformaldehyde/Phosphate-Buffered Saline (PBS) (pH 7.4) for fixation. The brains were then post-fixed with 4% paraformaldehyde for one night at 4 ℃ and then were stored in 30% sucrose/PB for two days at 4 ℃.

## Immunohistochemistry

Brains were frozen with methylbutane placed on dry ice at around −30 °C. Then the brains were sliced in 30 µm thick coronal sections serially distributed in eight wells at −20 °C in a cryoprotective solution containing 20% glycerol and 30% Ethylene glycol in 0.05 M PB (pH 7.4). Brain sections were first washed in 0.1 M PBS with 0.4% Triton X-100 (Sigma-Aldrich) to remove the cryoprotectant. Sections were then incubated in 0.3% $H_2O_2$ for 1 hour to quench endogenous per-oxidase activity, then washed 3*10 min in 0.1 M PBS with 0.4% Triton X-100. Different wells of brain sections were used to perform the different immunofluorescence protocols. They were all incubated with anti-c-Fos rat antibodies (1:50,000, c-Fos antibody – 226 017; Synaptic System), and incubated with either anti-Goat-Orexin (1:5,000, Goat monoclonal IgG, sc-80263 Santa Cruz) or anti-Rabbit-TH (1:40,000, Rabbit polyclonal, 213 102 Synaptic System) for 48 h at 4 °C in 0.1 M PBS containing 0.4% Triton X-100. After being washed 3*10 min in 0.1 M PBS with 0.4% Triton X-100, sections were then incubated for 3 hours at room temperature in 0.1 M PBS with 0.4% Triton X-100 containing biotinylated rabbit Anti-Rat IgG antibody diluted to 1:, (Vector Laboratories), and donkey anti-goat IgG 647 nm antibody diluted to 1:500 (Invitro-gen, A21447) for Orexin, or donkey anti-rabbit IgG 647 nm antibody diluted to 1:500 (Invitrogen, A31573) for TH. Then, they were washed 3*10 min with 0.1 M PBS with 0.4% Triton X-100. Following an incubation of 1h30 with streptavidin (SA)-HRP (Alexa Fluor Tyramide SuperBoost Kit, streptavidin; Life Technologies, 1:1000) in 0.1 M PBS with 0.4% Triton X-100, sections were washed 3*10 min in 0.1 M PBS with 0.4% Triton X-100, and incubated for 10 min in Alexa Fluor 488-conjugated Tyramide (Molecular Probes, Eugene, OR, USA) by diluting the stock solution 1:500 in 0.0015% $H_2O_2$/ amplification buffer. The reaction was terminated after 10 min by rinsing the tissue in 0.1 M PBS. Following 3 washes with Phosphate-Buffered Saline Tween (PBST) buffer, sections were mounted, dried, and cover-slipped with prolonged Gold anti-fading reagent containing 4′,6-diamidino-2-phenylindole (DAPI) (Molecular Probes, Eugene, OR) and stored at 4 ℃. Sections were then imaged using an Axioscan Z.1 slide scanning microscope (Zeiss, Germany). Images were collected with a ×20 objective (N.A. Plan-Apochromat 20×/0.8 M27 Air/0.8.) and a 0.45 Orca Flash camera. Dapi, c-Fos (Alexa-488), tdTomato (tdT), TH or hypocretin/orexin (Orx) (Alexa-647) were acquired using appropriate filter cubes, according to manufacturer recommendations. Exposure time was defined by signal distribution across gray values (>20% of measure gray values on 16 bits images). For all sections, five mosaics were acquired (5 µm steps) then projected on a single plane, using the depth of focus "wavelet" function of the Zen software (version 3.1). Images were not modified and were directly imported in Neuroinfo (MBF Bioscience, USA) for quantification.

## Cell counting

The Allen Brain Reference Atlas (Adult Mouse) was used as reference for all structures. In triple-stained fluorescent sections, drawings of structures and automatic plotting of cFos+, tdTomato+, and Tyrosine Hydroxylase (TH) or Orexin/ hypocretin (Orx) neurons was then made using a computerized image analysis system (Neuroinfo software, MBF Bioscience, USA). The cFos+, tdTomato+, double (cFos+/tdTomato+), and triple labeled neurons were automatically plotted and counted with the Neuroinfo software in all mice on 10 sections per mice with approximately 1 mm spacing from the forebrain to the medulla oblongata. Cells were considered labeled when they exhibited clear cytoplasmic (for tdTomato, TH or Orx) or nuclear (for cFos) staining. Four representative mice per group (Sol-Mod; $n = 4$, Mod-Sol; $n = 4$, NW-Sol; $n = 4$, Sol-NW; $n = 4$) were analyzed. There was no statistical difference between hemispheres (not shown) and, therefore, counting from the left and right hemispheres were summed.

## Statistical analysis of sleep–wake parameter and tdT and cFos expression

Statistical analyses were conducted using Matlab (Mathworks, USA) or Python. Non-parametric tests (Kruskal-Wallis followed by Mann-Whitney) were used to compare sleep-wake quantities after drug administration or NW protocol. Pearson's correlation was employed to assess the association between the expression levels of tdT and cFos in individual animals. Data is organized by the following macroregions: cortex, telencephalon, thalamus, hypothalamus, hippocampal formation, mesencephalon, pons, and medulla.

## Statistical analysis of specific activation in the Sol condition

To detect brain structures with different activation patterns across conditions, we calculated the densities (cells per unit area) of tdt+ and cFos+ neurons in Sol, Mod, Nwday, and Nwnight conditions. Because densities are positive and right-skewed, we used generalized linear models (Gamma family, log link) with robust (HC0) standard errors, testing directional contrasts of Sol > Mod, Sol > Nwday, and Sol > Nwnight (animal = experimental unit). To leverage the TRAP design, we also computed the reactivation index (RI), defined as (number of double-labeled cFostdt neurons)/(number of tdt+ neurons), and analyzed it with a binomial GLM (logit link, HC0 Ses) using frequency weights equal to tdT counts. A structure was classified as more activated in Sol if it satisfied both density criteria (tdT and cFos each higher in Sol than the comparators) and the RI criteria RI(Mod–Sol)> RI(Sol–Mod) at $p < 0.05$, with RI(Nwday–Sol)> RI(Sol–Nwday) accepted as trend-level evidence when $0.05 \leq p < 0.10$ (all tests directional in the "greater" sense).

## Statistical analysis of specific activation in the Mod, Nwday, or Nwnight conditions

A structure was classified as more activated in Mod if it met three criteria. First, the tdt+ neuron density in Mod exceeded that in Sol and Nwday, as tested with generalized linear models (Gamma family, log link) using robust (HC0) standard errors, one-sided in the "greater" direction with the animal as the experimental unit. Second, the cFos+ neuron density in Mod exceeded that in Sol, Nwday, and Nwnight under the same model and directional contrasts (note that tdt is not acquired in Nwnight, so only cFos is compared there). Third, leveraging the TRAP design, the reactivation index (RI = # cFostdT/ # tdT) was greater in Sol–Mod than in Mod–Sol, tested with a binomial GLM (logit link, HC0 Ses) and frequency weights equal to tdt counts (animals with tdt = 0 were excluded from RI). Only structures satisfying all three criteria were labeled Mod-specific.

A structure was classified as more activated in Nwday if two density criteria and one RI criterion were met. First, the tdt+ neuron density in Nwday exceeded that in Mod and Sol, tested with Gamma-family GLMs (log link) using robust (HC0) standard errors, one-sided in the "greater" direction (animal = experimental unit). Second, the cFos+ neuron density in Nwday exceeded that in Nwnight, Mod, and Sol under the same model and directional contrasts. Third, leveraging TRAP, the reactivation index (RI = # cFostdt/ # tdt) was greater in Sol–Nwday than in Nwday–Sol,

assessed with a binomial GLM (logit link, HC0 Ses) and frequency weights equal to tdT counts (animals with tdT = 0 were excluded from RI).

A structure was considered specifically activated in NWnight if the cFos+ neuron density in NWnight exceeded that in Nwday, Mod, and Sol, tested with Gamma-family GLMs (log link) with robust (HC0) Ses, one-sided in the "greater" direction. (Because tdt is not acquired in Nwnight, the RI criterion is not evaluated for this group).

### Statistical analysis of the correlation between the densities of tdT and cFos neurons and total locomotion

For each structure and modality, we quantified monotonic associations between density and locomotion using Spearman's $\rho$ (two-sided tests). To control multiplicity across structures, Benjamini–Hochberg FDR correction was applied separately to the p-values from tdT and from cFos, yielding q-values (q_Tdt, q_cFos). A structure was deemed "correlated after FDR" if q_Tdt < 0.05 and q_cFos < 0.05, with concordant signs of $\rho$ (both positive or both negative). Analyses were performed in Python (pandas, scipy, statsmodels), with verification of animal–condition matching and area conversions.

### Statistical analysis of specific activation of the TH-expressing neurons

Activation of TH-expressing neurons was quantified using three measures: the proportion of TH+ neurons co-expressing tdT (tdT/TH), the proportion co-expressing cFos (cFos/TH), and a triple-labeling reactivation index measuring the probability that TdT-tagged TH+ neurons were reactivated (Fos+) during the cFos-tagged episode. Statistical analyses were performed using binomial generalized linear models (GLMs) applied directly to the raw cell counts. For TdT/TH and cFos/TH, the model used the number of double-labeled neurons as "successes" and the remaining TH+ neurons as "failures." For triple labeling, "successes" corresponded to Fos + TdT + TH+ neurons and "failures" to TdT + TH+ neurons lacking Fos, restricting analyses to biologically valid condition pairs (Mod↔Sol and Nwday↔Sol), as Nwnight lacks a corresponding TdT episode.

Condition was included as a categorical predictor, with Sol set as the reference when present, yielding odds ratios and 95% confidence intervals. All *p*-values were corrected for multiple testing using the Benjamini–Hochberg false discovery rate procedure. Ratios shown in figures are provided for visualization only, whereas statistical inference relies entirely on the binomial GLM, which incorporates the appropriate denominators (TH+ or tdT + TH+) and accounts for variability in sampling across animals.

### Correlation analysis between locomotor activity and TH neuron activation

To assess the relationship between behavioral activity and activation of TH-positive neurons, we performed correlation analyses between total locomotion and the number of TH neurons labeled with either TdTomato (TH–tdT) or cFos (TH–cFos). For each animal, total locomotion was quantified separately for TdT and cFos. Correlations were computed independently for TH–tdT and TH–cFos counts using Spearman's rank correlation coefficient, chosen for its robustness to non-normal data distributions and small sample sizes. Analyses were performed separately for each brain structure. *P*-values were corrected for multiple comparisons using the Benjamini–Hochberg false discovery rate (FDR) procedure, applied independently to TH–tdT and TH–cFos correlations across structures. Structures were considered significantly correlated with locomotion only if correlations were significant after FDR correction for both TH–tdT and TH–cFos and showed the same direction of effect.

### Statistical analysis of specific activation of the Orx-expressing neuron

In the lateral hypothalamic area, we quantified (i) the proportion of orexin neurons co-expressing tdT (tdT/Orx), (ii) the proportion co-expressing cFos (cFos/Orx), and (iii) a triple-labeling reactivation measure defined as Fos + tdT + Orx divided by tdT + Orx neurons. Statistical inference relied on binomial generalized linear models (logit link) applied

directly to raw counts. For tdT/Orx and cFos/Orx, "successes" were double-labeled neurons (tdT + Orx or Fos + Orx) and "failures" were Orx neurons lacking the marker, using the total Orx count as the number of trials. For triple labeling, "successes" were Fos + tdT + Orx neurons and "failures" were tdT + Orx neurons lacking Fos, thus conditioning reactivation on the tdT-labeled Orx population. Triple-labeling contrasts were restricted to biologically valid paired conditions (Mod-Sol and Nwday-Sol), whereas Nwnight (without tdT) was only included in cFos-based analyses. For the triple-labeling, comparisons were performed separately within each pair (Mod-Sol versus Sol-Mod; Nwday-Sol versus Sol-Nwday). P-values were corrected for multiple testing using the Benjamini–Hochberg false discovery rate procedure across the planned contrasts.

## Supporting information

**S1 Fig. Sleep scoring.** Time spent in wakefulness in the first **(a)** and second **(b)** experimental session per 30 min, starting at the injection of solriamfetol (Sol), modafinil (Mod) or the beginning of the induction of NW, open field). Nonparametric Kruskal-Wallis followed by post hoc Mann-Whitney test, * $p < 0.05$.
(TIF)

**S2 Fig. Correlation between the expression of tdTomato and cFos in the circadian experiment.** Scatter plots show log10-transformed tdT expression versus log10-transformed cFos expression for individual mice submitted to the circadian protocol. As shown for the previous groups, Nwday (tdT expression) and Nwnight (cFos expression) were highly correlated. Each dot represents one structure of a given macrostructure (color coded). The diagonal indicates the linear regression fit. Coefficients of determination ($R^2$) and corresponding p-values are shown in each panel, demonstrating a strong and highly significant positive correlation between tdT and cFos expression across all mice. Raw data underlying the Figure is shown in S3 Data.
(TIF)

**S3 Fig. Sol-specific structures without robust locomotion–density correlations after FDR.** Panels illustrating the five Sol-specific regions that did not show a significant correlation between locomotion and neuronal density across both markers after Benjamini–Hochberg FDR ($q < 0.05$) with concordant signs. For each structure, left panel: Tdt density versus locomotion; right panel: cFos density versus locomotion. Points are individual animals (Tdt in red; cFos in green). Black line: ordinary-least-squares fit; shaded band: 95% CI (for visualization only). Panel titles report Spearman's $\rho$, uncorrected $p$, and FDR-adjusted $q$. Densities were computed as counts/area ($mm^2$). Non-selection reflects either lack of FDR significance in one modality and/or opposite correlation directions between Tdt and cFos. Raw data underlying the Figure is shown in S1 and S3 Data.
(TIFF)

**S4 Fig. Relationship between locomotor activity and orexin neuron activation.** Scatter plots illustrate the relationship between total locomotion and the number of orexin (Orx) neurons labeled with TdTomato (Orx–tdT, left panel, red dots) or expressing cFos (Orx–cFos, right panel, green dots). Each dot represents one animal. Black lines indicate linear regression fits shown for visualization purposes only. Spearman's rank correlation coefficient ($\rho$) and associated p-values are indicated in each panel. Locomotor activity was not significantly correlated with Orx–tdT neuron counts, whereas a significant positive correlation was observed between locomotion and Orx–cFos labeling, partly suggesting increased acute recruitment of Orx neurons in animals with higher locomotor activity Raw data underlying the Figure is shown in S1 and S5 Data.
(TIFF)

**S1 Table. Densities of tdTomato-positive (tdt+) and cFos+ neurons in Sol-specific structures across conditions.** For each of the nine structures classified as Sol-specific, the table reports density (cells per unit area; see Methods) of

tdt+ and cFos+ neurons in Sol, Mod, NWday, and Nwnight as mean ± SEM, with n (animals) shown beneath each value. (Note: for Nwnight, only cFos is available; tdt is not acquired in that condition. Within each structure and marker (tdt, cFos), densities were modeled with generalized linear models (Gamma family, log link) using the animal as the experimental unit and robust (HC0) standard errors. The table lists planned directional contrasts testing Sol > Mod, Sol > Nwday, and Sol > Nwnight (where applicable), reported as one-sided Wald $p$-values. $P$-values are unadjusted and correspond to these targeted, pre-specified comparisons. Raw data underlying the Figure is shown in S3 Data.
(DOCX)

**S2 Table. Reactivation Index (RI) across groups for structures activated by Sol condition.** For each structure, the table reports the RI = (cFostdt neurons)/tdt neurons) as mean ± SEM for the four paired-design groups Mod-Sol, Sol-Mod, NWday-Sol, Sol-Nwday (n is the number of animals contributing to RI for each structure).Planned between-group RI comparisons are shown as two-sided Wald $p$-values from a binomial GLM (logit link) with robust (HC0) standard errors and frequency weights = tdt counts: Mod-Sol versus Sol-Mod, Nwday-Sol versus Sol-Nwday, Mod-Sol versus Nwday-Sol, and Sol-Mod versus Sol-Nwday. Numeric $p$-values are reported. Raw data underlying the Figure is shown in S3 Data.
(DOCX)

**S3 Table. TH structures (all results).** Binomial GLMs were fitted on raw counts. For TdT_TH and cFos_TH contrasts ('X vs Sol'), OR < 1 indicates a higher proportion in Sol than in X; OR > 1 indicates a lower proportion in Sol. For triple reactivation, the contrast tests Sol_second versus Sol_first within each mirror pair. Q-FDR values are Benjamini–Hochberg corrected within each analysis family. Raw data underlying the Figure is shown in S4 Data.
(DOCX)

**S4 Table. Orexin neurons (LHA): Summary of planned statistical tests.** Odds ratios (OR) with 95% confidence intervals are reported from binomial GLMs. For contrasts of the form 'X vs Sol', OR < 1 indicates higher labeling in Sol. For triple reactivation tests, OR > 1 indicates higher reactivation when Sol is the second episode. $P$-values are FDR-corrected across all planned tests. Raw data underlying the Figure is shown in S5 Data.
(DOCX)

**S5 Table. Listing of the structures showing a high reactivation index in all groups.** Structures with averaged reactivation index above 0.3 in all groups and with a $p > 0.05$ in one-way ANOVA (d.f. = 3,12), excepting the Lateral Reticular Nucleus showing just a value just above (0.0051). The numbers in each group show the mean ± SEM. Raw data underlying the Figure is shown in S3 Data.
(DOCX)

**S1 Data. Excel spreadsheet showing raw spontaneous locomotor activity data (distance traveled, in meters) recorded during a continuous 2-hour session in each experimental conditions.** Each row corresponds to an individual animal (name in A). The total distance traveled is provided for both sessions (Condition 1, tdT staining and Condition 2, cFos staining). Data shown in Fig 2.
(XLS)

**S2 Data. Excel spreadsheet showing electrophysiological quantification of spectral parameters derived from power spectrum density (PSD) analysis.** Each row corresponds to an individual animal (name in A). Delta band power and peak frequency, as well as theta band power and peak frequency, were calculated for three 10-minute recording epochs: immediately after injection (epoch 1), ~1 hour post-administration (epoch 2), and ~2 hours post-administration (epoch 3) during Condition 1. Data shown in Fig 3.
(XLS)

**S3 Data. Excel spreadsheet showing for each structure (A) in condition 1 (tdT) and condition 2 (cFos) for each animal (D) the number of cFos labeled neurons (E), tdT neurons (F), cFostdT double-labeled neurons (G) and surface of the structure (area in mm$^2$).** Data shown in Figs 4–12).
(XLSX)

**S4 Data. Excel spreadsheet showing for each structure (A) in condition 1 (tdT) and condition 2 (cFos) for each animal (D) the number of TH (E), cFosTH labeled neurons (F), tdTTH neurons (G) and cFostdTTH triple-labeled neurons (H).** Data shown in Figs 11–13.
(XLSX)

**S5 Data. Excel spreadsheet showing for the lateral hypothalamic area (A) in condition 1 (tdT) and condition 2 (cFos) for each animal (D) the number of Orx (E), cFosOrx labeled neurons (F), tdTOrx neurons (G) and cFostdTOrx triple-labeled neurons (H).** Data shown in Fig 14.
(XLSX)

## Acknowledgments

We thank the CIQLE platform of the SFR Santé Lyon Est for scanning the sections.

## Author contributions

**Conceptualization:** Claudio Marcos Queiroz, Pierre-Hervé Luppi.

**Data curation:** Renato Maciel, Justin Malcey, Amarine Chancel, Blandine Duval, Pierre-Hervé Luppi.

**Formal analysis:** Justin Malcey, Kylian Gautier, Amarine Chancel, Théo Brunel, Claudio Marcos Queiroz, Pierre-Hervé Luppi.

**Funding acquisition:** Pierre-Hervé Luppi.

**Investigation:** Justin Malcey, Claudio Marcos Queiroz, Pierre-Hervé Luppi.

**Methodology:** Justin Malcey, Blandine Duval, Claudio Marcos Queiroz, Pierre-Hervé Luppi.

**Project administration:** Claudio Marcos Queiroz, Pierre-Hervé Luppi.

**Supervision:** Patrice Fort, Claudio Marcos Queiroz, Pierre-Hervé Luppi.

**Validation:** Kylian Gautier, Claudio Marcos Queiroz, Pierre-Hervé Luppi.

**Visualization:** Justin Malcey, Kylian Gautier, Amarine Chancel, Claudio Marcos Queiroz.

**Writing – original draft:** Pierre-Hervé Luppi.

**Writing – review & editing:** Claudio Marcos Queiroz, Pierre-Hervé Luppi.

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
