## [Editor Report · Decision Letter 0]

7 Jan 2026

Dear Pierre-Hervé,

Thank you for submitting your revised manuscript entitled "Drug-induced versus non-pharmacological wakefulness: similar or different states? A whole brain analysis in TRAP2 transgenic mice" for consideration as a by PLOS Biology.

Your manuscript has now been evaluated by the PLOS Biology editorial staff, and I am writing to let you know that we would like to send your submission back to the original reviewers.

Once your full submission is complete, your paper will undergo a series of checks in preparation for peer review. After your manuscript has passed the checks it will be sent out for review. To provide the metadata for your submission, please Login to Editorial Manager (https://www.editorialmanager.com/pbiology) within two working days, i.e. by Jan 09 2026 11:59PM.

Kind regards,

Luke

Lucas Smith, Ph.D.

Senior Editor

PLOS Biology

lsmith@plos.org

---

## [Decision Letter · Decision Letter 1]

27 Feb 2026

Dear Pierre-Hervé,

Thank you for your patience while we considered your revised manuscript "Drug-induced versus non-pharmacological wakefulness: similar or different states? A whole brain analysis in TRAP2 transgenic mice" for publication as a Methods and Resources Article at PLOS Biology. Please note that I am currently handling your manuscript on behalf of my colleague Luke Smith since he is out of the office this week. I am also very sorry for the delays that you have experienced during this round of the peer review process. This revised version of your manuscript has been evaluated by the PLOS Biology editors, the Academic Editor and the original reviewers.

Based on the reviews, I am pleased to say that we are likely to accept this manuscript for publication, provided you satisfactorily address the remaining points raised by the reviewers. After discussions with the Academic Editor, we will not make the request from Reviewer #1 to include additional positive control data essential for the revision. In addition, I would be grateful if you could please make sure to address the following editorial and data-related requests pasted below (A-H):

(A) We routinely suggest changes to titles to ensure maximum accessibility for a broad, non-specialist readership. In this case, we would suggest a minor edit to the title, as follows. Please ensure you change both the manuscript file and the online submission system, as they need to match for final acceptance:

“Pharmacological and non-pharmacological methods of inducing wakefulness activate distinct neural populations in the mouse brain"

(B) In the manuscript file, we note that the Acknowledgements section contains funding information. Please move this information to the Funding Disclosure section.

(C) In the conflict of interest statement in the online submission system, I would be grateful if you could please provide the following statement given your role as an Academic Editor for the journal:

“I have read the journal’s policy and the authors of this manuscript have the following competing interests: PHV is a member of PLOS Biology’s Editorial Board."

(D) You may be aware of the PLOS Data Policy, which requires that all data be made available without restriction: http://journals.plos.org/plosbiology/s/data-availability. For more information, please also see this editorial: http://dx.doi.org/10.1371/journal.pbio.1001797

-Supplementary files (e.g., excel). Please ensure that all data files are uploaded as 'Supporting Information' and are invariably referred to (in the manuscript, figure legends, and the Description field when uploading your files) using the following format verbatim: S1 Data, S2 Data, etc. Multiple panels of a single or even several figures can be included as multiple sheets in one excel file that is saved using exactly the following convention: S1_Data.xlsx (using an underscore).

-Deposition in a publicly available repository. Please also provide the accession code or a reviewer link so that we may view your data before publication.

Figure 2B-C, 3B-C, 4, 5A-B, 6A-B, 7D-E, 8D-E, 9D-E, 10, 11D-G, 12D-G, 13, 14D-F, S1A-B, S2, S3, S4

(E) Please also ensure that each of the relevant figure legends in your manuscript include information on *WHERE THE UNDERLYING DATA CAN BE FOUND*, and ensure your supplemental data file/s has a legend.

(F) Please ensure that your Data Statement in the submission system accurately describes where your data can be found and is in final format, as it will be published as written there.

(G) Per journal policy, if you have generated any custom code during the course of this investigation, please make it available without restrictions. Please ensure that the code is sufficiently well documented and reusable, and that your Data Statement in the Editorial Manager submission system accurately describes where your code can be found. More information on our Code Policy, what and how to share can be found here: https://journals.plos.org/plosbiology/s/code-availability

(H) Please ensure that you are using best practice for statistical reporting and data presentation. These are our guidelines https://journals.plos.org/plosbiology/s/best-practices-in-research-reporting#loc-statistical-reporting and a useful resource on data presentation https://journals.plos.org/plosbiology/article?id=10.1371/journal.pbio.1002128

- If you are reporting experiments where n ≤ 5, please plot each individual data point.

We expect to receive your revised manuscript within two weeks.

*Published Peer Review History*

*Press*

Best regards,

Richard

Richard Hodge, PhD

rhodge@plos.org

On behalf of:

Lucas Smith, PhD

lsmith@plos.org

Reviewer remarks:

Reviewer #1: The authors now show behavioral data during wake induced in the different conditions. This perhaps shows a correlate of the differences in activation. However, I am disappointed that my major concern - lack of an essential positive control (something critical to the interpretation of any of the findings) was not addressed. It is unclear why the authors feel that they should heavily weigh differences between tdT and cFos as being critically important for characterizing the different types of waking, when they have no data indicating how much these signals overlap for the same, repeated condition (eg. sol-sol). This is an essential question that requires answering in order to fairly interpret any of the findings of the paper. Since the authors chose not to address it, the study continues to suffer a serious flaw.

Reviewer #2: The authors have addressed my concerns, I have no further questions.

Reviewer #3: authors properly addressed the concerns.

Reviewer #4: This study systematically compares whole-brain neural activation patterns between drug-induced and natural wakefulness using the TRAP2 transgenic mouse model. The revised manuscript has effectively addressed all previous concerns and presents strong innovation, supported by comprehensive and robust data. Minor revisions are suggested to further enhance its quality, as follows:

1. The comparison with Hasan's 2009 study remains somewhat superficial, as it currently focuses solely on differences in dosage and analytical approaches. It is recommended to incorporate direct comparisons of key findings—including co-activated brain regions and the extent of overlap among differentially activated areas—and to more clearly elaborate on the methodological and interpretative advances of the present work relative to this prior study.

2. The discussion section does not adequately address the unexpected observation of "insufficient activation of the classical arousal system." To deepen the depth and expand the scope of the discussion, it would be valuable to explore potential explanations in the context of existing literature, such as the specificity of drug dosage, variations in animal model characteristics, or distinctions between different subtypes of arousal states.

---

## [Editor Report · Decision Letter 2]

9 Mar 2026

Dear Pierre-Hervé,

Thank you for the submission of your revised Research Article "Pharmacological and non-pharmacological methods of inducing wakefulness activate distinct neural populations in the mouse brain" for publication in PLOS Biology and thank you for addressing the last reviewer and editorial requests in this revision. On behalf of my colleagues and the Academic Editor, Paul J Shaw, I am pleased to say that we can in principle accept your manuscript for publication, provided you address any remaining formatting and reporting issues. These will be detailed in an email you should receive within 2-3 business days from our colleagues in the journal operations team; no action is required from you until then. Please note that we will not be able to formally accept your manuscript and schedule it for publication until you have completed any requested changes.

**IMPORTANT: Note, that after discussing your paper further with the team, we ended up feeling it is better suited for publication as a 'Research Article' rather than a 'Methods and Resources Article'. I have therefore taken the liberty of switching your article type back to Research Article (note this will *not* require any other changes to the manuscript). I apologize for the back and forth here, as I realize I was the one who suggested you change the article type to 'Resource' in the first place and please let me know if you would like to discuss this further.

PRESS

Sincerely,

Luke

Lucas Smith, Ph.D.

Senior Editor

PLOS Biology

lsmith@plos.org